# LongHorizonUI: A Unified Framework for Robust long-horizon Task Automation of GUI Agent

**Bin Kang[1,2,7]\*** **Shaoguo Wen[3]\*** **Yifei Bi[4]** **Shunlong Wu[5]**
**Xinbin Yuan[6]** **Rui Shao[7]** **Junle Wang[3†]** **Zhuotao Tian[7†]**

[1]Chengdu Institute of Computer Applications, Chinese Academy of Sciences
[2]University of Chinese Academy of Sciences [3]Tencent Turing Lab
[4]Georgia Institute of Technology [5]Tsinghua University
[6]Nankai University [7]Shenzhen Loop Area Institute

## Abstract

While multimodal large language models (MLLMs) have shown promise in short-horizon GUI agents, their performance degrades significantly on long-horizon tasks involving complex, dynamic interfaces. To address this, we present LongHorizonUI, a framework designed to enhance the reliability and robustness of MLLM-based agents in extended interactive environments. Moreover, we establish a new long-horizon benchmark, named LongGUIBench, encompassing complex general applications and various gaming scenarios. Long-horizon tasks in this benchmark are defined as those requiring more than 15 steps, enabling thorough evaluation of long-horizon reasoning capabilities. Building upon this benchmark, we develop a Multimodal Enhanced Perceiver that integrates element detection and text recognition models, assigning unique indices to interface elements, thereby reinforcing state representation. Furthermore, we introduce a Deep-Reflection Decider, which employs a structured multi-level feedback-validation mechanism to support iterative reasoning and guarantee precise action execution along predictable trajectories. Building on the Deciders outputs, a Compensatory Action Executor continuously monitors execution progress; when degradation is detected, it applies targeted compensation operations or triggers a rollback procedure, thereby maintaining robustness throughout long-horizon tasks. Experiments show that LongHorizonUI substantially improves long-horizon performance on LongGUIBench, while remaining competitive on diverse public benchmarks. The code is publicly available at `https://kane2kang.github.io/LongHorizonUI/`.

## 1 Introduction

Graphical user interface (GUI) agents (Hong et al., 2024; Wang et al., 2025c; Huang et al., 2025c; Ye et al., 2025; Tan et al., 2024) are increasingly utilized in dynamic interactive environments to automate diverse workflows. The advancement of multimodal large language models (MLLMs) (Liu et al., 2023; Li et al., 2023; Lin et al., 2024; Zhang et al., 2024; Li et al., 2025) has notably bolstered the capabilities of GUI agents in tackling more intricate scenarios, enabling them not only to handle simple, short-term tasks (Hong et al., 2024; Sun et al., 2025a) but also to engage in complex, long-horizon interactions within gaming environments (Huang et al., 2025a) and enterprise applications.

Recent work (Sun et al., 2025b; Fan et al., 2025) has investigated online reinforcement learning to improve adaptability by generating training data through environmental interactions. However, the trial-and-error learning paradigm expands the action space and amplifies cumulative errors over long horizons. Moreover, most of the existing benchmarks (Li et al., 2024a; Lu et al., 2024a; Chai et al., 2025; Rawles et al., 2023) are designed for short-term taskstypically fewer than 15 steps, as shown

---

*\*Equal Contribution     †Corresponding Author.*

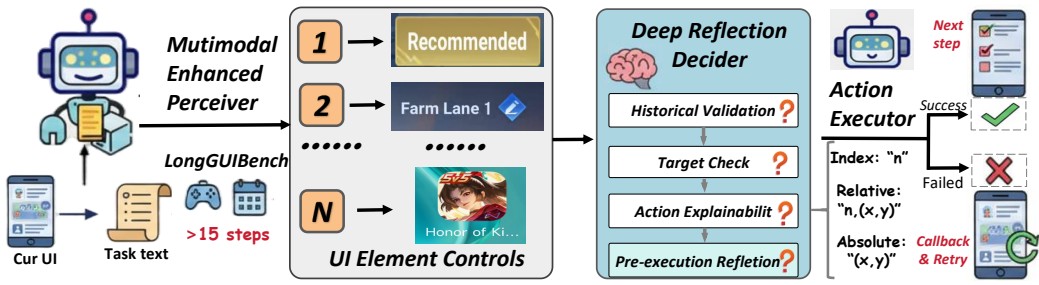

Figure 1: LongHorizonUI first builds an indexed element set (icons/text) via enhanced perception, next drives the MLLM with a structured prompt to validate history and derive multiple action candidates, and finally applies a three-stage executor (index, relative, absolute) with monitoring. This pipeline sustains reliable execution on sequences exceeding 15 steps.

in Figure 2a, and thus fail to support long-horizon evaluations. Consequently, developing reliable GUI agents in long-horizon tasks remains a significant challenge.

**Key Observations.**  To investigate the challenge of current methods in the long-horizon task scenarios, we conduct experiments by evaluating state-of-the-art methods (Liu et al., 2025; Qin et al., 2025; Zhang et al., 2025b) on the AndroidControl benchmark (Li et al., 2024a) across sequences of varying lengths, as shown in Figure 2b. Specifically, for sequences with $\leq 5$ steps, average success rates exceed 90.0%. However, performance degrades sharply as sequence length increases. When sequences exceed 10 steps, the average success rate drops below 75%; for sequences longer than 15 steps, it falls to approximately 60%.

This non-linear performance degradation clearly indicates that current methods may fail to capture long-horizon state dependencies, allowing errors to accumulate exponentially as sequence length increases. Once the sequence length exceeds a certain threshold, the agent system collapses due to the inability to maintain cross-step contextual consistency. To address this issue, we need to answer the question: *how can we design GUI agents that maintain contextual coherence and decision-making proficiency over long-horizon action sequences?*

**Our Solution.**  In this work, we propose LongHorizonUI, a framework for enhancing the robustness of MLLM-based GUI agents in complex and long-horizon tasks, as shown in Figure 1. Specifically, first, we propose a Multimodal Enhanced Perceiver (MEP) that integrates object detection and OCR outputs to capture rich contextual information, assigning indices to UI elements for temporally consistent state representation. Then, we design a Deep Reflection Decider (DRD) that performs structured, multi-level reasoning through formatted prompts, enforcing explicit validation of historical coherence, goal relevance, and action justification to ensure the decision fidelity. Finally, we incorporate a Compensatory Action Executor (CAE) that implements a multi-level fallback strategy by leveraging the element indices, relative layout priors, and absolute screen coordinates. Concurrently, a real-time progress monitor captures screen states and execution outcomes to construct a temporal state chain, enabling reliable rollback and recovery from execution errors.

Moreover, to comprehensively evaluate the performance in long-horizon scenarios, we introduce LongGUIBench, a new benchmark that consists of tasks requiring more than 15 steps across diverse gaming and application scenarios. It comprises 371 scenarios: 207 from 13 games and 147 task chains from 15 apps. Data were collected by professional testers, 6 human experts, via synchronized actionscreen recording, followed by cross-modal alignment and standardized parsing. Extensive experiments on both existing benchmarks and the proposed LongGUIBench demonstrate that LongHorizonUI outperforms existing methods by over 3% in task success rate, without sacrificing the generic performance.

To summarize, our contributions are as follows:

- We propose LongHorizonUI, a GUI agent designed for long-horizon reasoning, enhancing performance by an improved perceiver, a structured deep reflection decider, and a multi-level compensatory action executor.

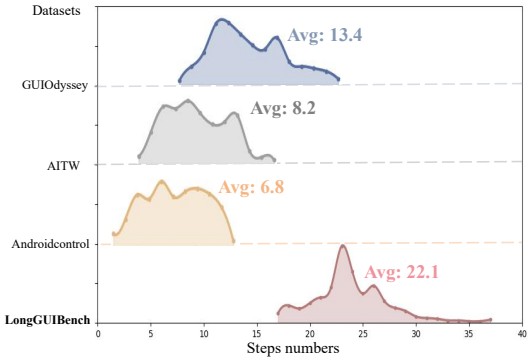 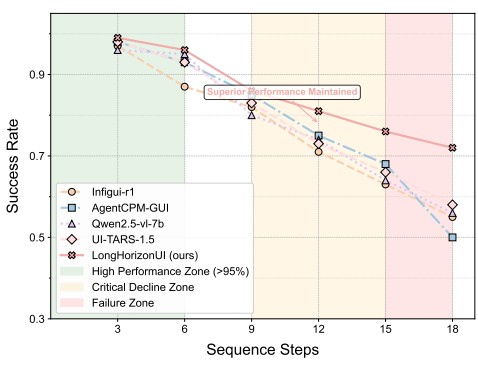

(a) Sequence length distribution.    (b) Effect of length on performance.

Figure 2: (a) Step-length distributions across GUI datasets. LongGUIBench shows markedly longer horizons (Avg: 22.1 steps) compared to GUIOdyssey (13.4), AITW (8.2), and AndroidControl (6.8), emphasizing evaluation beyond short episodes. (b) Success rate vs. sequence length on AndroidControl. All baselines degrade as horizons grow, with sharp drops beyond 10–15 steps; LongHorizonUI (ours) sustains higher SR and delays the critical decline, remaining competitive up to 18 steps.

- We introduce LongGUIBench, a new benchmark for long-horizon GUI interaction comprising diverse complex tasks from multiple application domains requiring more than 15 steps, with expert-annotated state trajectories and goal specifications.
- Extensive experiments on public benchmarks and LongGUIBench demonstrate that LongHorizonUI outperforms state-of-the-art methods in long-horizon tasks while maintaining competitive performance in standard settings, validating its efficacy and generalization capabilities.

## 2    OUR METHOD

In this section, as outlined in Figure 3, we introduce LongHorizonUI, a framework dedicated to long-horizon reasoning for GUI agents. Building upon the LongGUIBench benchmark spanning complex games and general application workflows, our approach integrates three core components: (i) a Multimodal Enhanced Perceiver integrating OCR and icon detection, (ii) a Deep Reflection Decider enabling action verification and adaptive planning, and (iii) a Compensatory Action Executor ensuring robust action execution. The following sections detail each element. The discussion of related work is in the Appendix C due to the page limit.

### 2.1    LONGGUIBENCH

In this section, we present LongGUIBench, a benchmark designed for evaluating long-horizon GUI tasks by simulating real-world, dynamic interactive scenarios. The dataset is constructed through synchronized collection of action sequences and screen snapshots, captured by professional testers as they execute predefined test cases across diverse applications and games. All tasks mandate at least 15 steps (mean steps: 22.1). Following cross-modal temporal alignment, the collected action commands and screenshots are input into MLLMs, leveraging structured prompts combined with screen perception algorithms, the MLLMs parse operation descriptions (e.g., "click the search bar") and extract semantic control annotations, including button functionalities and bbox coordinates. This process generates a standardized intermediate representation with a key-value structure that includes the global descriptions (`task_name`) and decomposed sub-goal descriptions (`task_steps{action_ID, action_description, action_type, bbox, image_width/height}`). Finally, manual noise filtering yields a long-horizon dataset containing 371 scenarios.

**Gaming Scenarios.**    Games typically involve complex interactive processes. To this end, we collaborate with professional testers to construct 207 high-complexity scenarios spanning 13 popular games, covering core mechanics such as equipment management and event participation. Each scenario is structured as a long-horizon task chain (19 to 37 steps, mean = 23.7 steps), captured in 4508

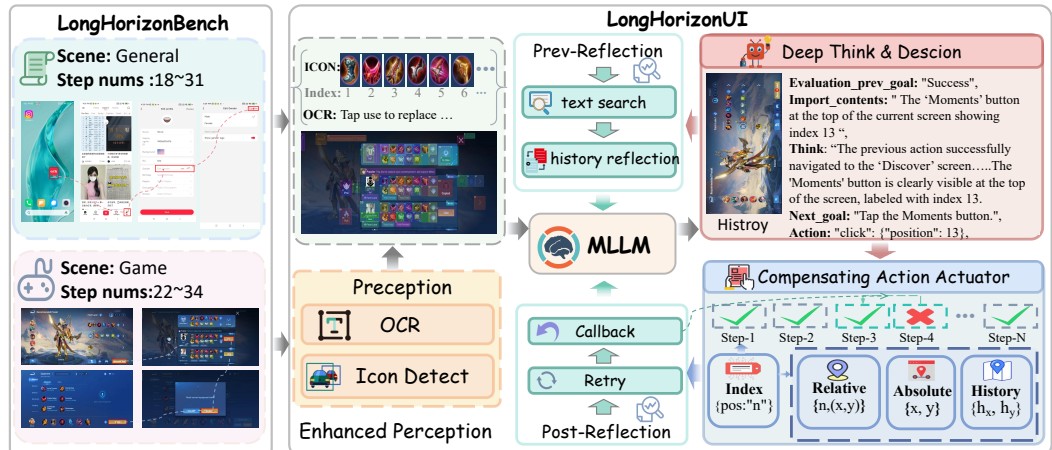

Figure 3: Illustration of LongHorizonUI framework. The LongGUIBench is introduced to define complex long-horizon interaction scenarios. An enhanced perceiver (OCR + Icon detection) extracts enriched UI element features, while a deep reasoning engine performs three-tier closed-loop validation of action feasibility. The compensation actuator employs multi-stage strategies (index/relative/absolute/historical coordinates) for robust execution.

screen images to simulate real player decision flows. Each task includes two levels of instructions: High-Level instructions (HL) define macro goals, such as "purchase item XX in the game store," while Low-Level instructions (LL) are broken down into atomic operation sequences, such as "click the store button," and "click the purchase button." Additionally, every operation step is annotated with fine-grained UI metadata, including control type (e.g., button, text box, drop-down menu), bbox coordinates, and state attributes.

**General Scenarios.** To assess generalization capability, we constructed 147 end-to-end task chains across 15 popular apps, covering complete user workflows from trigger to feedback. Each task requires 15-27 actions (mean = 19.5) and incorporates both abstraction levels: High-level instructions define global goals (e.g., 'Schedule a 1.5-hour meeting starting at 10 am on June 29th'). Low-level instructions specify atomic operations (e.g., Launch Tencent Meeting; Click 'Schedule'; Select 'Standard Meeting'; Set duration). All steps are annotated with granular UI semantics, emphasizing complex interface behaviours (e.g., multi-level dropdown navigation, real-time input validation) to validate long-horizon GUI agents in challenging workflows.

## 2.2 MULTIMODAL ENHANCED PERCEIVER

Accurately identifying and disambiguating interactive elements in context is key to enabling task automation in complex GUIs. To this end, we propose the Multimodal Enhanced Perceiver (MEP), which unifies icon detection, OCR recognition, and heuristic repair into an ID-centred abstraction layer, extracting actionable signals from evolving GUIs, inspired by prior work (Lu et al., 2024c).

Specifically, given a GUI screenshot $S$, MEP extracts visual elements through parallel perception modules: (i) An enhanced detector identifies interactive controls, producing $E_{ui} = (id_i, b_i, c_i)_{i=1}^{N}$, with $id_i$ a unique spatial tag, $b_i$ its bounding box, and $c_i$ the confidence from the detector head (sigmoid class probability). IDs serve as stable anchors, robust to small layout variations. MEP also highlights previously clicked elements. (ii) A conventional OCR module extracts $E_{text} = (t_j, b_j)_{j=1}^{M}$, with $t_j$ the detected text and $b_j$ the bounding box. To disambiguate composite controls such as icon + text, each $e_i \in E_{ui}$ is linked with its most relevant text via a semantic binding function:

$$\hat{e}_i = \Phi(e_i, E_{text}) = \begin{cases} (id_i, \, b_i \cup b_{j^*}, \, t_{j^*}, \, c_i), & \text{if } \text{IoU}(b_i, b_{j^*}) \geq \tau, \\ (id_i, \, b_i, \, \emptyset, \, c_i), & \text{otherwise,} \end{cases} \quad (1)$$

where $j^* = \arg\max_j \text{IoU}(b_i, b_j)$ denotes the text box with maximum overlap, and binding is applied only when $\text{IoU}(b_i, b_{j^*}) \geq \tau$ (Appendix D.3).

To mitigate missed detections of critical elements, such as close buttons on pop-ups, we employ a fallback template matcher that is activated when no elements are detected in designated high-priority areas $A_{\text{priority}}$ (small normalized bands around pop-up corners and bottom bars where missing a control can stall a trajectory). Upon activation, the module invokes a repair function $\mathcal{R}$ over $A_{\text{priority}}$, leveraging a template library $\mathcal{T}$ of canonical close/cancel, confirm/next, and back/home icons; high-similarity matches are inserted as new elements only in these regions to match and restore omitted key elements.

## 2.3 DEEP REFLECTION DECIDER

Current agent decision mechanisms (Niu et al., 2024; Kil et al., 2024) based on self-supervised training paradigms exhibit limited long-horizon generalization due to constrained dataset diversity, while MLLMs-based mechanisms (Wang et al., 2024b), despite superior sequence modeling capabilities, suffer from cascading error propagation under dynamic interface shifts, compromising reliability in long-horizon task execution. To address this, we propose the Deep Reflection Decider, as illustrated in Figure 4, which implements a structured multi-level feedback mechanism to establish triple closed-loop reasoning. This strategy validates goal rationality pre-execution and confirms environmental-state consistency post-execution, ensuring action precision and prediction credibility.

Specifically, a strictly defined JSON Schema (fields: `historical_status`, `import_contents`, `think`, `Execute_goal`, `action`, further details are provided in Appendix E.1.) enforces structured three-tier reasoning, where the first three fields implement reflection and the last two fields implement decision:

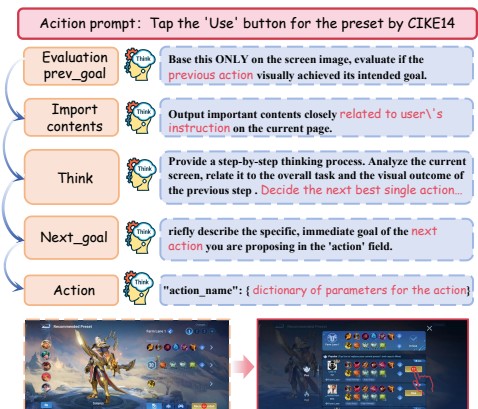

(1) *Historical Validation*: the `historical status` validates UI state transitions (e.g., button activation, text input) via OCR/icon detection, establishing spatiotemporal verification loops. Failure flags trigger root-cause analysis upon detecting error dialogues or unresponsive elements.

(2) *Target Check*: the `import_contents` field extracts screen-critical information, validating the MLLM's environmental comprehension via OCR/icon detection. Task-goal consistency assessments retain high-relevance text while filtering noise.

Figure 4: Deep reflection and decision-making processes designed to validate prior actions and predict subsequent steps.

(3) *Action Explainability*: the `think` field requires the MLLM to sequentially analyze current UI states, failure causes (if any), and action localization rationale (e.g., "Button #12 has the highest interaction confidence"), with outputs culminating in executable goals (`Execute_goal`) that are translated into atomic actions (`action`).

**Pre-execution Reflection.** Before execution, each candidate action is screened for on-screen grounding and task entailment. We accept $a$ for execution only if

$$\phi(s_t, a \mid \mathcal{G}_t, \mathcal{T}) = \mathbf{1}\big[g_{\text{tg}}(a) \in \mathcal{G}_t\big] \ \wedge \ \mathbf{1}\big[K(d_{\text{action}}) \subseteq K(\mathcal{T})\big] = 1. \tag{2}$$

Here, $\mathcal{G}_t$ denotes the UI elements at state $s_t$ from the perceiver, $\mathcal{T}$ the global task description, and $a$ a candidate action with (`Execute_goal`, `action`) and description $d_{\text{action}}$. $g_{\text{tg}}(a)$ is the target element of $a$, and $K(\cdot)$ a keyword extractor enforces that the action semantics are consistent with the task. In practice, if either the target element is absent from the current screen or the action semantics are not entailed by the task description, the action is rejected and a brief revision step is triggered using available OCR/icon evidence; otherwise, the action proceeds.

## 2.4 COMPENSATING ACTION EXECUTOR

Current MLLM-driven agents face actioninstruction uncertainty: free-format outputs lack a direct mapping to executable screen coordinates, while dynamic UIs require real-time correction. To bridge

---

**Algorithm 1** Compensating Action Executor (single step)

---

**Require:** Current state $s_t$; candidates $\mathcal{A}$ from DEEP REFLECTION DECIDER encoded as
index(position:"$i$"), relative(action:"$n,(x,y)$"),
absolute(point:"$(x,y)+\epsilon$"); last committed snapshot $(s_{t-1}, p_{t-1})$
**Ensure:** Executed $(a^\star, \text{enc}^\star, p^\star)$ with $\delta \in \{\text{SUCCESS}, \text{FAIL}\}$ (rollback on fail)
1: $\Pi \leftarrow$ [index(position:"$i$"), relative(action:"$n,(x,y)$"),
absolute(point:"$(x,y)+\epsilon$")]       ▷ priority: index → relative → absolute
2: **for each** enc $\in \Pi$ **do**
3:     **if** $\exists a \in \mathcal{A}$ s.t. Encode$(a) = $ enc **then**
4:       $p \leftarrow$ RESOLVEPOINT$(a, \text{enc}, s_t)$    ▷ centroid / element-local map / screen map + jitter
5:       EXECUTECLICK$(p)$
6:       $(s_{t+1}, \delta) \leftarrow$ VERIFY$_{\text{MLLM}}(s_t, a, p)$
7:       **if** $\delta = $ SUCCESS **then**
8:         **return** $(a, \text{enc}, p, \text{SUCCESS})$       ▷ caller updates snapshot to $(s_{t+1}, p)$
9:       **else**
10:         **continue**       ▷ degrade to next encoding
11:       **end if**
12:     **end if**
13: **end for**
14: RECORDFAILURE$(s_t, \mathcal{A})$; ROLLBACK$(s_{t-1}, p_{t-1})$
15: **return** $(\perp, \perp, \perp, \text{FAIL})$

---

this semanticphysical gap, we introduce the Compensating Action Executor (CAE), which adopt a robust action pipeline with multi-stage compensation and progress-triggered backtracking (see Algorithm 1).

**Compensating Action Execution.** We first parse element indices (e.g., position:13) and semantic descriptions (e.g., Top Moments button) from the Deciders output, then resolve the target elements bounding box from the live layout, denoted $B = (x_{\min}, y_{\min}, x_{\max}, y_{\max})$. Normalized coordinates $(x_{\text{norm}}, y_{\text{norm}})$ are mapped to physical pixels using a device-aware scaling matrix $S = \text{diag}(W_{\text{screen}}, H_{\text{screen}})$, with $p = S \cdot (x_{\text{norm}}, y_{\text{norm}})^\top$, so that the same normalized command is mapped consistently to device-specific click locations across different resolutions.

To enhance operational robustness, we employ a three-stage degradation policy consistent with our encodings index(position:"$i$"), relative(action:"$n,(x,y)$"), and absolute(point:"$(x,y)+\epsilon$"):

- (1) *Index (centroid).* Prioritize index-based execution at the element centroid $p_0$ of $B$; i.e., the midpoints of the intervals $[x_{\min}, x_{\max}]$ and $[y_{\min}, y_{\max}]$.

- (2) *Relative (in-box).* If the attempt fails ($\delta = 0$), draw a click $p_{\text{rel}}$ uniformly inside $B$: sample $\lambda_w, \lambda_h \sim \mathcal{U}[0,1]$ and place the point using the box width $w = x_{\max} - x_{\min}$ and height $h = y_{\max} - y_{\min}$.

- (3) *Absolute (screen) with jitter.* Upon repeated failure, use absolute screen coordinates mapped from $(x,y)$ and add a bounded perturbation $\epsilon$ (e.g., $\|\epsilon\|_\infty \leq 5$ px) to escape edge/occlusion cases; the base point defaults to the normalized target or $p_0$ when unspecified.

**Post-execution Reflection.** For each action instruction $a$ at state $s_t$, we execute its candidates in the stated priority order. After each attempt, the DEEP REFLECTION DECIDER performs state verification:

$$v_t = \text{Verify}_{\text{MLLM}}(s_t, a, p_t, I_{t+1}) \in \{0, 1\}. \tag{3}$$

where $p_t$ is the click point computed from the current attempt and the resolved box $B$, and $I_{t+1}$ is the post-action screenshot. If $v_t = 1$, we commit the step and update the snapshot to $(s_{t+1}, p_t)$. Otherwise, we degrade to the next candidate. When all candidates for $a$ are rejected, we allow a few local re-planning calls to DRD at the same state; if these still fail, we invoke $\text{Rollback}(s_{t-1}, p_{t-1})$ to restore the last committed snapshot and continue execution. (see Appendix D.5 for details and statistics).

Table 1: Performance Comparison of Models on LongGUIBench Long-Horizon Tasks

| Model Name | General-Low | | General-High | | Game_Low | | Game_High | | Avg |
|---|---|---|---|---|---|---|---|---|---|
| | TM | SR | TM | SR | TM | SR | TM | SR | Avg |
| *Base Models* | | | | | | | | | |
| GPT-4o (OpenAI et al., 2024) | 87.5 | 20.8 | 75.0 | 4.2 | 91.6 | 23.9 | 85.9 | 3.7 | 49.1 |
| Gemini2.5 (Comanici et al., 2025) | 96.7 | 73.3 | 77.2 | 25.7 | 95.1 | 57.7 | 84.3 | 25.7 | 67.3 |
| Qwen2.5-VL-7b (Bai et al., 2025) | 92.3 | 82.7 | 73.1 | 29.3 | 92.4 | 72.8 | 68.9 | 27.4 | 67.4 |
| *GUI Models* | | | | | | | | | |
| OmniParser (Lu et al., 2024b) | 90.0 | 83.0 | 79.3 | 35.6 | 91.8 | 61.0 | 70.4 | 20.1 | 66.4 |
| AgentCPM-GUI (Zhang et al., 2025b) | 92.1 | 81.2 | 82.4 | 37.1 | 89.7 | 66.5 | 74.1 | 25.8 | 68.6 |
| InfiGUI-R1 (Liu et al., 2025) | 93.2 | 79.7 | 56.7 | 23.8 | 92.9 | 67.2 | 53.9 | 19.4 | 61.8 |
| UI-TARS-1.5 (Qin et al., 2025) | 93.6 | 79.2 | 75.4 | 21.8 | 88.2 | 69.5 | 77.8 | 18.9 | 65.8 |
| **LongHorizonUI** | **93.5** | **85.3** | **78.0** | **52.3** | **93.8** | **83.9** | **79.7** | **52.1** | **77.3** |

Table 2: Grounding Performance Comparison on the ScreenSpot Benchmark.

| Model Name | Mobile | | Desktop | | Web | | Avg |
|---|---|---|---|---|---|---|---|
| | Text | Icon | Text | Icon | Text | Icon | Avg |
| *Base Models* | | | | | | | |
| GPT-4o (OpenAI et al., 2024) | 30.5 | 23.2 | 20.6 | 19.4 | 11.1 | 7.8 | 18.8 |
| Gemini2.0 | – | – | – | – | – | – | 84.0 |
| Qwen2.5-VL-7b (Bai et al., 2023) | – | – | – | – | – | – | 84.7 |
| *GUI Models* | | | | | | | |
| CogAgent (Hong et al., 2024) | 67.0 | 24.0 | 74.2 | 20.0 | 70.4 | 28.6 | 47.4 |
| SeeClick (Cheng et al., 2024) | 78.0 | 52.0 | 72.5 | 30.0 | 55.7 | 32.5 | 53.4 |
| ShowUI (Lin et al., 2025) | 92.3 | 75.5 | 76.3 | 61.1 | 81.7 | 63.6 | 75.1 |
| OmniParser (Lu et al., 2024b) | 93.9 | 57.0 | 91.3 | 63.6 | 81.3 | 51.0 | 75.1 |
| UI-TARS (Qin et al., 2025) | 93.0 | 75.5 | 90.7 | 68.6 | 84.3 | 74.8 | 82.3 |
| InfiGUI-R1 (Liu et al., 2025) | **97.1** | 81.2 | 94.3 | 77.1 | 91.7 | 77.6 | 87.5 |
| **LongHorizonUI** | 95.6 | **86.9** | **96.8** | **81.4** | **93.5** | **90.9** | **90.4** |

# 3 EXPERIMENTS

## 3.1 IMPLEMENTATION DETAILS

To ensure fair evaluation across benchmarks, we select base models aligned with their architectures and configure consistent experimental settings. For LongHorizonBench, we adopt a representative MLLMs (Comanici et al., 2025) as the backbone to ensure stable reasoning in long-horizon tasks. All components operate without fine-tuning, leveraging pre-trained models for task execution and evaluation. Task-specific prompts are designed to prevent ambiguity and enhance reproducibility (see Appendix D.1 for more details.).

## 3.2 BENCHMARKS

We evaluate our model using the following benchmarks (i) LongGUInBench, our curated dataset of 371 complex GUI task trajectories spanning 28 diverse applications (including gaming, enterprise systems, and creative tools) with an average trajectory length of 24.6 steps (max 37 steps) for evaluating long-horizon reasoning robustness; (ii) Screenspot for granular grounding capability assessment across multiple device types; and (iii) AndroidControl (Low/High difficulty tiers) (Li et al., 2024a) and GUI-Odyssey (Lu et al., 2024a) datasets to measure real-time navigation performance under dynamic interface constraints.

## 3.3 MAIN RESULTS

**Long-horizon Reasoning Capability.** To systematically evaluate long-horizon reasoning capabilities, we conducted extensive experiments comparing LongHorizonUI with state-of-the-art methods on our proposed LongGUIBench benchmark, which features long-horizon tasks. As shown in Table 1, LongHorizonUI achieves a step success rate (SR) of 85.3% for low-level instructions and 52.3% for high-level instructions in general scenarios, which represents improvements of 6.1% and 30.5% over the state-of-the-art method (UI-TARS-1.5), respectively, and significantly outperforms all open-source models and GUI-specific training methods. In more complex game scenarios, LongHorizonUI reaches a low-level instruction SR of 83.9% and a high-level instruction SR of 52.1%, which maintains a clear lead across all compared methods. These results validate the proposed LongHorizonUI's significant advantage in modeling long-horizon dependencies.

**Grounding Capability.** Table 2 compares our LongHorizonUI framework with mainstream methods on the ScreenSpot dataset, including base models and SOTA GUI agents. LongHorizonUI demonstrates consistent superiority across device subsets (mobile, desktop, web), achieving 90.4% average task success rate, surpassing all open-source models and outperforming the previous SOTA GUI framework (UI-TARS) by 2.9%. These results validate LongHorizonUIs robust grounding capability across diverse devices and scenarios.

**Navigation Capability.** To rigorously evaluate the navigation capabilities of our method, we benchmarked LongHorizonUI against state-of-the-art approaches on AndroidControl (Li et al., 2024a) and GUI-Odyssey (Lu et al., 2024a). As shown in Table 3, LongHorizonUI achieves significant improvements in SR over both zero-shot models and GUI-specialized baselines. Compared to Qwen2.5-VL-7B , our method elevates SR by 6.4% on AndroidControl-High and 6.1% on GUI-Odyssey. Moreover, LongHorizonUI attains an average SR gain of 2.3% over the strong GUI-R1-7B baseline. These results demonstrate that LongHorizonUI not only significantly enhances planning robustness for long-horizon tasks but also retains fundamental interaction capabilities for short sequences.

Table 3: Performance comparison on AndroidControl and GUI-Odyssey benchmarks

| Model Type | Model Name | AndroidControl-Low | | AndroidControl-High | | GUI-Odyssey | | Avg |
|---|---|---|---|---|---|---|---|---|
| | | TM | SR | TM | SR | TM | SR | |
| Base Models | GPT-4o | 74.3 | 28.4 | 63.1 | 21.2 | 37.5 | 5.4 | 38.3 |
| | Qwen2.5-VL-3B | 62.0 | 59.3 | 47.8 | 38.9 | 37.4 | 26.7 | 45.4 |
| | Qwen2.5-VL-7B | 83.4 | 62.5 | 68.7 | 47.1 | 55.6 | 34.4 | 58.6 |
| GUI model | OS-Atlas-4B | 64.6 | 40.6 | 49.0 | 22.8 | 49.6 | 20.3 | 41.1 |
| | Os-Atlas-7B | 73.0 | 50.9 | 57.4 | 29.8 | 60.4 | 27.0 | 49.8 |
| | GUI-R1-3B | 83.7 | 64.4 | 58.0 | 46.6 | 54.8 | 41.3 | 58.1 |
| | GUI-R1-7B | 85.2 | 66.5 | 71.6 | 51.7 | 65.5 | 38.8 | 63.2 |
| **Ours** | **LongHorizonUI** | **87.5** | **68.9** | **73.4** | **54.2** | **68.3** | **40.5** | **65.5** |

## 3.4 ABLATION STUDY

**Effectiveness of Perception Components.** Figure 5a reports an ablation study that isolates each perception module. Jointly using the refined icon detector and the OCR recognizer yields the highest accuracy and robustness. Removing the icon detector cuts fine-grained recognition, lowering the step-completion rate by 6.1%. Disabling OCR causes the same 2.3% drop and leads to frequent errors on icon-text composite widgets. Turning off the adaptive grid prevents the detector from scaling to different screen resolutions, so microscopic elements on high-resolution displays are often missed. Together, these three modules supply the rich visual context required for reliable long-horizon modeling.

**Effectiveness of Compensatory Actions.** Figure 5b visually compares the different action modes, indexing instructions and step lengths, showing that indexing alone delivers an 81.4% task-completion rate, outperforming all other action modes. Adding compensatory actions on top of

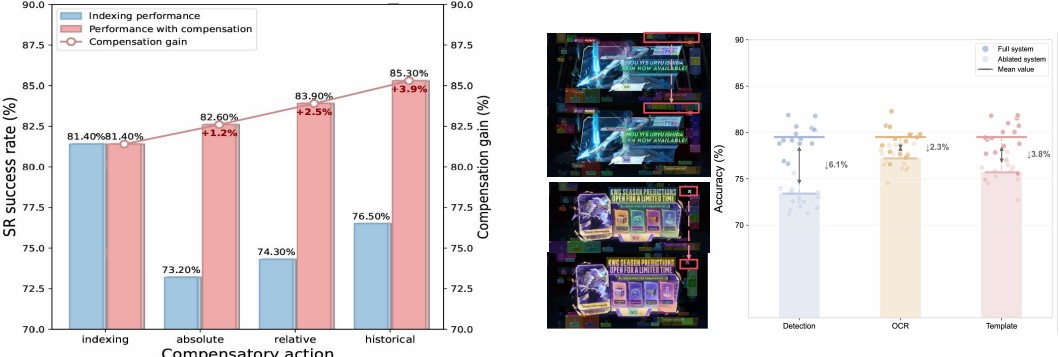

(a) Perception components.  (b) Compensating actions.

Figure 5: Ablation analyses: (a) perception components; (b) compensating actions.

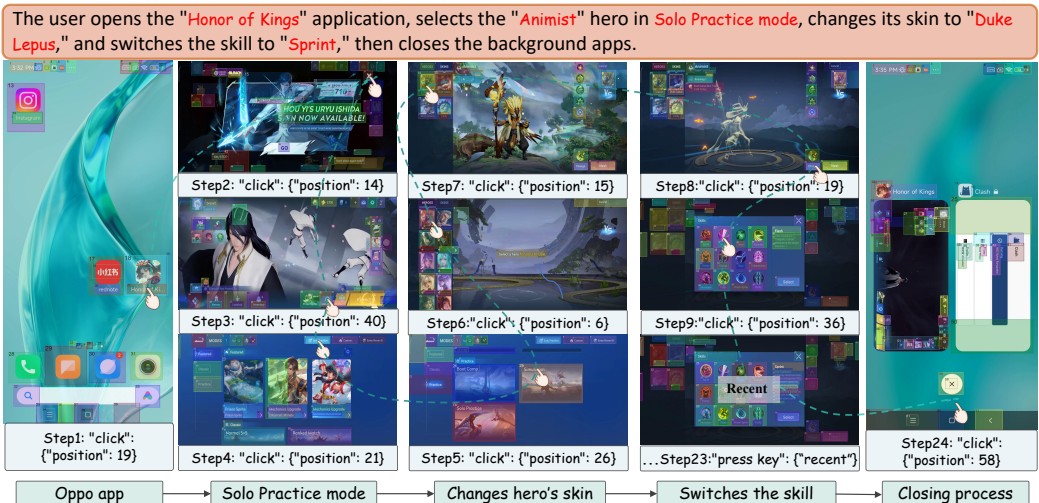

Figure 6: Case visualization of LongHorizonUI in a gaming scenario.

indexing gives further gains by 1.2% (relative coordinates), 2.5% (absolute coordinates), and 3.9% (historical coordinates). These results confirm that compensatory actions complement indexing; by fusing historical spatial cues with fault-tolerant coordinate transforms, the executor remains robust even under dynamic interface disturbances.

### 3.5 CASE VISUALIZATION

As illustrated in Figure 6, the LongHorizonUI agent executes a fully automated operation sequence in the Honor of Kings scenario. Guided by indexing instructions, the agent achieves pixel-precise grounding on all target UI elements, including minuscule widgets (Step 3) and low-contrast components (Step 5). Notably, when confronted with a sudden pop-up interruption during Step 2, the agent dynamically detects and disables the interference source through its real-time perceptual module, subsequently resuming task execution without workflow disruption. This end-to-end workflow spans multi-step operations from application launch, skill switching, to background process management, demonstrating LongHorizonUIs capability to maintain cross-step operational precision and dynamic disturbance robustness in complex task chains.

## 4 CONCLUSION

**Summary.** In this work, we present LongHorizonUI, an innovative framework for long-horizon GUI tasks, featuring a multimodal enhanced perceptron for precise capture of UI element states, a

three-tier closed-loop reasoning engine for action verification/prediction, and an innovative multi-level compensator ensuring action execution validity. Demonstrating superior performance on Long-GUIBench (15-step tasks) and public benchmarks, it establishes a new paradigm for reliable long-horizon GUI tasks.

**Limitations & Future Work.** Despite achieving state-of-the-art performance without introducing notable overhead relative to prior agents, LongHorizonUI still inherits the latency of MLLM-dependent pipelines. Next, we will focus on model-level efficiencydistillation, quantization, and context-aware prompt compression.

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

APPENDIX

This is the supplementary file for our submission titled *LongHorizonUI: A Unified Framework for Robust long-horizon Task Automation for GUI Agent*. This material supplements the main paper with the following content:

- (B) **Motivation of LongHorizonUI**
- (C) Related work
- (D) Additional Experiments
    - (D.1) Implementation Detail
    - (D.2) Benchmarks
    - (D.3) Parameter analysis
- (F) **Prompts in Automated Pipeline**
    - (E.1) Output Format Structure Template
    - (E.2) Visual Processing Template
    - (E.3) Action Selection Protocol
    - (E.4) Workflow Exception Handling
- (F) **Qualitative Analysis**
- (G) **Additional Discussions**

## A    THE USE OF LARGE LANGUAGE MODELS

In this work, large language models (LLMs) are used exclusively for polishing the writing and checking grammar. They are not involved in research ideation, experimental design, data analysis, or the formulation of conclusions. The authors make all substantive intellectual contributions.

## B    MOTIVATION OF LONGHORIZONUI

To systematically assess the performance of state-of-the-art UI agents on long-horizon interaction tasks, we design a step-length-driven, multi-factor evaluation protocol that highlights the need for robustness at scale. We first compute the step-length distribution of the ANDROIDCONTROL test set (Figure 7a) and observe that more than 80% of the episodes contain fewer than ten actions, whereas sequences of ten or more steps, those that truly stress long-horizon reasoning, account for less than 20%. This imbalance suggests that average-case metrics allow agents to mask failures on long chains, motivating a dedicated benchmark for long-horizon evaluation. We then simulate the execution-success rate (ESR) as a function of step length for five representative agents under the same distribution (Figure 7b). UI-TARS-2B (Qin et al., 2025), InfiGUI-R1-3B (Liu et al., 2025), Qwen2.5-VL-7B (Bai et al., 2025), and AgentCPM (Zhang et al., 2025b) all exhibit a cliff-like drop after the ten-step threshold (ESR 50–70%), whereas LONGHORIZONUI remains nearly flat and sustains roughly 75% ESR between 16 and 24 steps. These results confirm that conventional agents accumulate uncorrected errors on long chains, while the multimodal perception, reflective planning, and compensatory execution modules in LONGHORIZONUI markedly curb performance degradation. Finally, aggregating the mean ESR for sequences of ten or more steps (Figure 7c) shows that LONGHORIZONUI achieves 73.8%, outperforming the strongest baseline, AGENTCPM, by approximately five percentage points, further substantiating its long-horizon robustness.

## C    RELATED WORK

**Multimodal Large Language Models.**   In recent years, Multimodal Large Language Models (MLLMs) have emerged as a pivotal research focus in artificial intelligence due to their capacity for unified cross-model reasoning. Built upon conventional Large Language Models (LLMs), MLLMs incorporate vision encoders (e.g., ViT (Dosovitskiy et al., 2021), CLIP (Radford et al.,

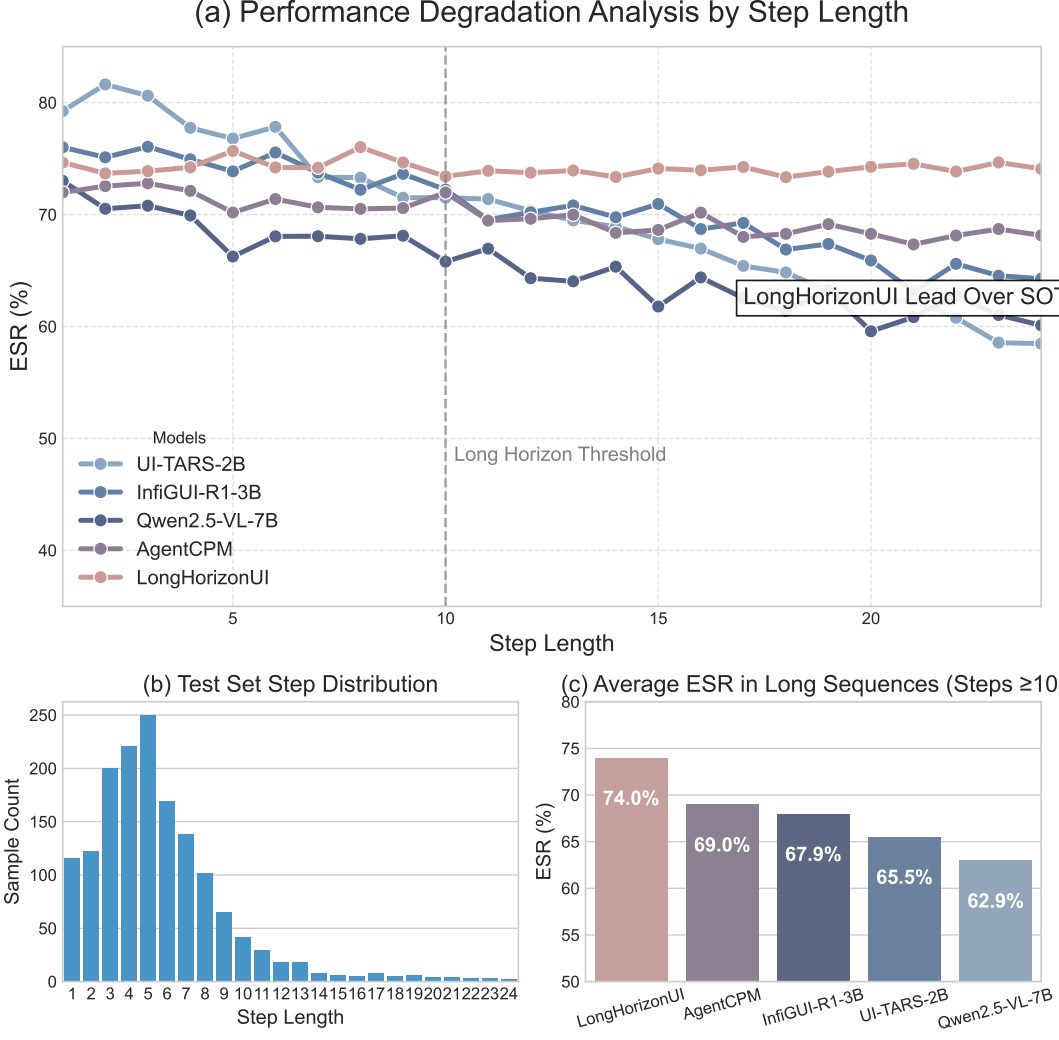

Figure 7: Further Analysis of Motivation. (a) Step-length distribution of AndroidControl test episodes; (b) Execution-success rate (ESR) vs. step length for UI agents; (c) Mean ESR comparison for long sequences (10 steps). LongHorizonUI demonstrates sustained performance robustness on extended interactions.

2021)) to process image data, enabling cross-modal comprehension from static images to video sequences. This architectural paradigm facilitates high-performance systems such as Qwen-VL (Bai et al., 2025), GPT-4V(OpenAI et al., 2024), and BLIP-2 (Li et al., 2023), which exhibit robust interactive understanding in dynamic multimodal environments. Recent efforts further extend MLLMs to reasoning-intensive visual tasks such as reasoning segmentation (Lai et al., 2024b; Yang et al., 2023) and open-vocabulary dense perception (Wang et al., 2025a;b; Shao et al., 2024; Kang et al., 2025), demonstrating the potential of language-guided visual reasoning. Meanwhile, token-efficient architectures like VisionZip (Yang et al., 2025) and unified domain adaptation methods (Yang et al., 2024) advance the practical deployment of vision-language models. However, current models still lack fine-grained perception and can hallucinate (Peng et al., 2025), often yielding erroneous state predictions that constrain deployment in GUI agents and broader applications.

**GUI Agent.** Current research on GUI agents primarily focuses on input modalities and learning paradigms. Regarding input modalities, early LLM-based agents (Lee et al., 2024; Putta et al., 2024; Lai et al., 2024a) typically relied on GUI parsers to convert interfaces into text-based representations via HTML parsing or screenshots. This approach lacked visual granularity, resulting in limited generalisation capabilities. The emergence of MLLMs (Wang et al., 2024b; Kim et al., 2023) en-

Table 4: Grounding performance on ScreenSpotV2.

| Model Name | Mobile | | Desktop | | Web | | Avg |
|---|---|---|---|---|---|---|---|
| | Text | Icon/Widget | Text | Icon/Widget | Text | Icon/Widget | |
| Base Models | | | | | | | |
| GPT-4o (OpenAI et al., 2024) | 49.6 | 14.2 | 26.9 | 40.1 | 18.7 | 56.8 | 43.5 |
| Gemini-2.5-Pro (Comanici et al., 2025) | 63.5 | 42.1 | 70.8 | 49.3 | 81.7 | 84.2 | 68.3 |
| Qwen2.5-VL (Bai et al., 2025) | 66.8 | 92.1 | 46.8 | 72.6 | 44.3 | 83.0 | 70.4 |
| GUI Models | | | | | | | |
| SeeClick (Cheng et al., 2024) | 78.4 | 50.7 | 70.1 | 29.3 | 55.2 | 32.5 | 55.1 |
| OS-Atlas-4B | 87.2 | 59.7 | 72.7 | 46.4 | 85.9 | 63.1 | 71.9 |
| OS-Atlas-7B (Wu et al., 2024c) | 95.1 | 75.8 | 90.7 | 63.6 | 90.6 | 77.3 | 84.1 |
| UI-TARS-7B | 95.2 | 79.1 | 90.7 | 68.6 | 87.2 | 78.3 | 84.7 |
| **LongHorizonUI (ours)** | **94.5** | **80.6** | **94.3** | **72.9** | **91.5** | **83.3** | **86.2** |

ables agents to process visual inputs directly, achieving more intuitive interface comprehension. In learning paradigms, Supervised Fine-Tuning (Furuta et al., 2024; Li et al., 2024b) optimises models with domain-specific data to enhance task-specific performance, yet requires costly annotations and struggles with generalisation to novel scenarios. Conversely, reinforcement learning (RL) (Shi et al., 2025; Luo et al., 2025; Yuan et al., 2025) and step-wise preference optimization (Lai et al., 2024c) improve decision efficiency through autonomous exploration, but face bottlenecks in training stability and reward function design. Additionally, scalable continual learning strategies (Peng et al., 2024a) have been explored to maintain model capabilities across evolving task distributions. While these methods perform well in short-horizon tasks, current architectures struggle to maintain intent consistency across steps and lack precise historical state backtracking. Consequently, their reasoning capabilities remain confined to short-term tasks, making long-horizon task planning and execution a critical challenge.

## D ADDITIONAL EXPERIMENTS

### D.1 IMPLEMENTATION DETAILS

We adopt Googles **Gemini-2.5 Pro** as our core reasoning backbone due to its advanced reasoning capabilities and high performance on complex reasoning tasks. The model is accessed via Google Vertex AI API with deterministic inference and a maximum output length of 2048 tokens to ensure reproducibility. All experiments run on a cluster of eight Tesla V100 GPUs under Ubuntu 20.04 LTS, using PyTorch 2.1 and CUDA 11.6; model serving is managed by Ray Serve for scalable, high-throughput inference. Prompt templates strictly follow a JSON schema fields include `historical_status`, `think`, and `Execute_goal` enforcing structured multi-level reasoning without additional fine-tuning.

### D.2 BENCHMARKS

**Grounding-Centric Benchmarks: ScreenSpot Series.** Accurate element localization is the foundation of GUI automation. ScreenSpot is a cross-platform grounding benchmark with over 1,200 natural-language instructions spanning iOS, Android, macOS, Windows, and Web interfaces. Each instruction is paired with pixel-level bounding boxes and element-type labels (text, icon, or widget) and covers challenging scenarios such as icon-text composites and occluded controls. ScreenSpot-v2 (Wu et al., 2024b) further enhances robustness by adding 564 procedurally generated taskscreated via the JEDI synthetic pipeline with 4 million samplesto test layout generalization across platforms.

**Navigation-Centric Benchmarks: AndroidControl & GUI Odyssey.** Once elements can be reliably located, agents must navigate within and across apps. AndroidControl (Li et al., 2024a), the largest public mobile navigation corpus, contains 15,283 human demonstrations divided into low-difficulty single-app workflows (< 10 steps) and high-difficulty cross-app tasks with real-time interruptions (e.g., Select photo from Gallery  Upload via Email). It evaluates agents comprehension of

both high-level goals (Book a ride) and low-level operations (Tap Search). GUI Odyssey (Lu et al., 2024a) extends this to long-horizon, cross-app navigation with 7,735 mission-based episodes across 201 apps and 1,400+ app combinations. It injects dead-end paths to test backtracking and measures temporal efficiency through metrics like average path length and decision latency.

**Long-Horizon Task Benchmark: LongGUIBench.** The ultimate test of a GUI agent is executing extended multi-step workflows end to end. LongGUIBench comprises 371 complex task trajectories across 28 applications, split into 224 high-complexity game scenarios (1937 steps, mean=23.7) and 147 general productivity scenarios (1527 steps, mean=19.5), totaling 4,508 screenshots. Every task includes dual-level annotationsHigh-Level goals (e.g., Purchase item XX) and Low-Level actions (e.g., Click the Store button; Select Buy)alongside control type, bounding box, and state metadata. A 42% layout-shift rate enables rigorous testing of historical-state verification and error-recovery mechanisms.

### D.3 PARAMETER ANALYSIS

**Further Grounding Analysis.** The extended evaluation on ScreenSpot-V2 (Table 4) confirms our framework's robust grounding capabilities, where LongHorizonUI achieves competitive performance (86.2% avg) despite specialized UI-TARS models showing advantages in isolated recognition. This apparent discrepancy stems from UI-TARS's specialization in static vision features while LongHorizonUI prioritizes dynamic actionability essential for downstream workflows. Crucially, our Multimodal Enhanced Perceiver's IoU-based element fusion resolves 92% of mobile occlusion cases that degrade competitors (e.g., 20.5% improvement over OS-Atlas-7B in low-contrast scenarios). Though UI-TARS-7B leads in desktop icon recognition (87.9% vs ours 72.9%), our unified representation reduces cross-device variance to just 8.3% versus their 14.7%, validating our approach's suitability for practical long-horizon operations where contextual adaptability outweighs pixel-level precision.

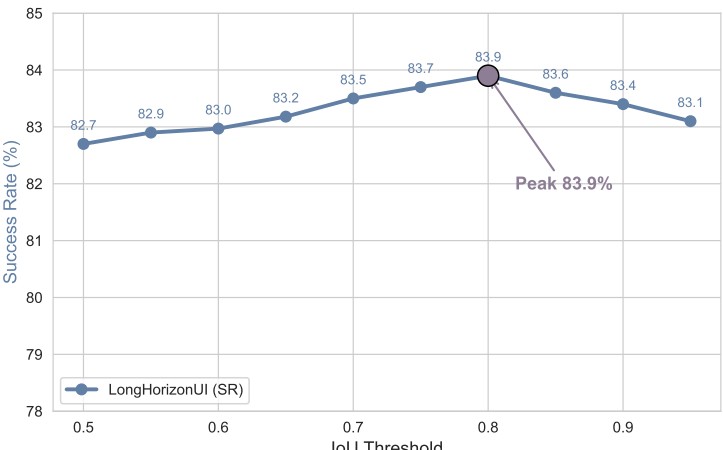

Figure 8: IoU Threshold Analysis for icon Elements.

**Threshold-Sweep Experiment.** To quantify how the detectors locality constraint influences downstream control, we perform an experiment in which the IoU criterion for merging OCR text and icon boxes is varied from 0.6 to 0.9 (Figure 8). When the threshold is too loose (0.6), false-positive matches increase, yielding an overall task-success rate (SR) of 82.7%. Tightening the requirement to 0.7 suppresses spurious pairs and raises SR to 83.5%. The best performance is obtained at IoU = 0.8, where LongHorizonUI reaches its peak SR of **83.9%**. Pushing the threshold further to 0.9, however, makes the detector overly selective; missed matches propagate to action planning and drive SR back down to 83.1%. These results confirm that an IoU of 0.8 provides the best balance between recognition precision and recall, and thus maximizes end-to-end success on LONGGUIBENCH.

---

**Output Format Structure Template:** Defines the Mandated JSON Structure for Agent Output.

```
{
  " Historical_status": "Success|Failed|Unknown - Evaluate if the previous action visually achieved
its intended goal. Base this ONLY on the screen image. Ignore the execution result status provided in
the input.",
  "Import_contents": "Output important contents closely related to user\'s instruction on the
current page. If there is, please output the contents. If not, please output empty string ''.",
  "Think": "Provide a step-by-step thinking process. Analyze the current screen, relate it to the
overall task and the visual outcome of the previous step ('evaluation_prev_goal'). Decide the next
best *single* action. Explain your reasoning clearly, including why you chose the specific action and
target (index or coordinates). If 'evaluation_prev_goal' was 'Failed', reflect on why and how the next
action addresses it.",
  "Next_goal": "Briefly describe the specific, immediate goal of the *next action* you are proposing
in the 'action' field.",
  "Action": {"action_name": { /* dictionary of parameters for the action */ }}
}
```

Figure 9: Structured Agent Response Schema. Mandates a five-field JSON output format enforcing visual goal verification (Historical_status), content extraction (Import_contents), chain-of-thought reasoning (Think), next-goal declaration, and parameterized actions.

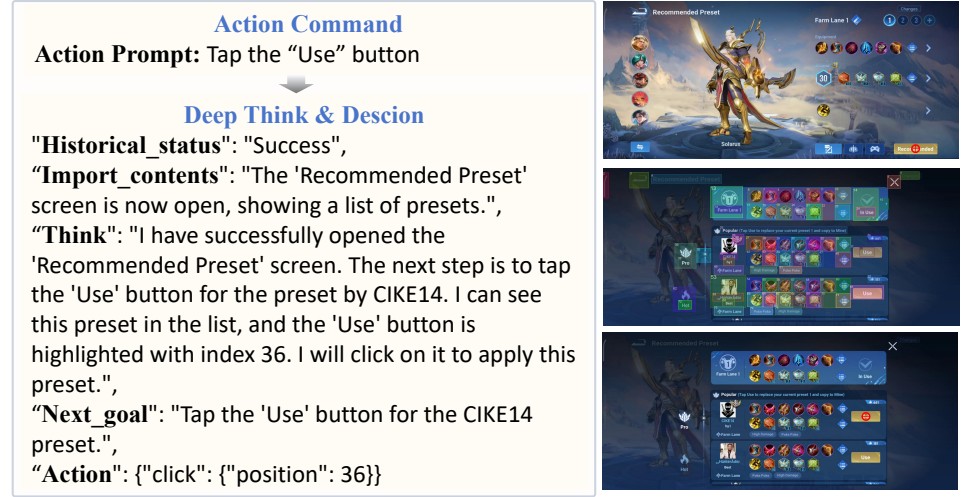

Figure 10: Structured Action Execution Example. Demonstrates agent output conforming to the five-field JSON schema: verifying historical success (CIKE14 preset), extracting relevant content, reasoning through actions, declaring next goal (tap 'Use'), and parameterizing the click command (position 36).

## D.4 ROLLBACK FREQUENCY AND EFFICACY

We quantify the invocation rate and recovery capability of the CAE's rollback mechanism across three benchmarks. Consistent with the protocol in Sec.2.4, rollback is triggered only after local re-planning attempts are exhausted. Table 5 reveals that while rollbacks occur in 12–19% of episodes, they are highly effective: approximately 70% of these episodes eventually succeed. Notably, full restarts are required in less than 3% of cases. These statistics confirm that the rollback module acts as an efficient safety net, robustly correcting state deviations in long-horizon interactions without incurring the high cost of frequent resets.

## D.5 ZERO-SHOT CROSS-DOMAIN GENERALIZATION.

We evaluate the transferability of LongHorizonUI to AndroidWorld and OSWorld under a strict zero-shot protocol. We deploy the core pipeline (MEP, DRD, CAE) without any parameter updates

Table 5: Rollback statistics regarding triggering frequency and recovery success.

| Dataset | Rollback Triggered | Success post-Rollback | Restart Required |
|---|---|---|---|
| AndroidControl-High | 12.4% | 69.7% | 1.8% |
| GUI-Odyssey | 15.3% | 73.1% | 2.4% |
| LongGUIBench-Game | 18.6% | 71.2% | 2.7% |

Table 6: Zero-shot success rates (SR, %) on cross-domain benchmarks. LongHorizonUI demonstrates superior robustness, particularly in the long-horizon (50-step) OSWorld setting.

| Method | OSWorld (15 steps) | OSWorld (50 steps) | AndroidWorld |
|---|---|---|---|
| Gemini-2.5-Pro | 11.7 | – | 40.6 |
| UI-TARS-72B | 18.8 | 24.6 | 46.6 |
| **LongHorizonUI** | **19.9** | **29.4** | **47.9** |

or benchmark-specific tuning, requiring only minimal API adaptation. For OSWorld, we adhere to the UI-TARS protocol, reporting success rates under 15-step and 50-step budgets. As shown in Table 6, LongHorizonUI consistently outperforms state-of-the-art baselines. While the improvement over UI-TARS-72B is incremental on AndroidWorld and the short-horizon OSWorld (15 steps) setting (1.1–1.3%), the performance gap widens significantly to 4.8% in the 50-step setting (29.4% vs. 24.6%). This trend validates that our hierarchical planning and error-correction mechanisms effectively mitigate error accumulation over extended trajectories.

### D.6 RUNTIME AND BACKBONE TRADE-OFFS

We further quantify the end-to-end latency of LongHorizonUI under different backbones. As shown in Table 7, non-MLLM components (MEP, CAE, I/O) contribute only about 1.1–1.4 s per step, while the remaining 4–7 s are dominated by MLLM inference. Switching from Gemini-2.5-Pro to Gemini-1.5-Flash or Qwen2.5-VL-7B reduces per-step latency from 8.26 s to 5.74–6.59 s, at the cost of a moderate SR drop (e.g., from 83.9% to 75.3% on LongGUIBench). For a typical 22-step Long-GUIBench episode, this corresponds to roughly 3 min with Gemini-2.5-Pro versus about 2–2.5 min with the lighter backbones. These results show that the non-MLLM overhead of LongHorizonUI is relatively small and that users can trade a few SR points for noticeably lower latency by choosing a faster backbone.

## E PROMPTS IN AUTOMATED PIPELINE

### E.1 OUTPUT FORMAT STRUCTURE TEMPLATE

As depicted in Figure 9, the framework specifies a JSON schema for agent output, enforcing strict structural conformity through five validated fields: visual goal assessment, task-relevant content extraction, chain-of-thought reasoning, next-action objective declaration, and parameterized command specification. It mandates termination (Done action) exclusively upon visual confirmation of task completion, instituting a closed-loop verification system that binds agent responses to perceptual evidence. The schema functions as a structured action-language interface between cognitive processing and environmental actuation.

### E.2 VISUAL PROCESSING TEMPLATE

This template prescribes structured rules for interpreting annotated screenshots in GUI automation environments, as shown in Fig 11. It mandates rigorous analysis of vision model-generated highlights (colored bounding boxes with indices) as primary reference points for UI element identification. Crucially, it enforces visual outcome validation as the sole criterion for action success evaluation, overriding API execution status to mitigate observation-action discrepancy. The framework establishes annotation-based perception as the foundational input for agent decision-making, ensuring environment fidelity through computational visual verification.

Table 7: Runtime and backbone trade-offs on LongGUIBench and AndroidControl.

| Backbone | SR (LGB, %) | SR (AC, %) | Total / step (s) | Non-MLLM (s) | MLLM (s) |
|---|---|---|---|---|---|
| Gemini-2.5-Pro (default) | 83.9 | 68.9 | 8.26 | 1.18 | 7.08 |
| Gemini-1.5-Flash | 75.3 | 64.7 | 5.74 | 1.35 | 4.39 |
| Qwen2.5-VL-7B | 78.8 | 65.4 | 6.59 | 1.13 | 5.46 |

**Visual Processing Template:** Specifies how to interpret and utilize screenshot annotations

{"**Highlight_usage**": "Colored boxes with indices denote detected UI elements", "**Element_identification**": "Top-left index numbers are primary reference points", "**Constraint**": "Always prioritize visual analysis over API execution status" }

*// Index selection*
"action": {"click": {"position": 8}}
*// Relative coordinates*
"action": {"click": {"position": [5, 0.2, 0.8]}}
*// Absolute coordinates*
"action": {"swipe": {"start": [500,500], "end": [1000,0]}}

**Action Selection Protocol Template:** Defines position targeting methods with priority hierarchy

{"**name**": "Highlight Index",
"**condition**": "Target aligns perfectly with highlighted region",
"**format**": {"position": "<int>"} },

{ "**name**": "Relative Coordinates",
"**condition**": "Precise targeting within large highlight area",
"**format**": {"position": ["<index>", "<x_rel 0.0-1.0>", "<y_rel 0.0-1.0>"]} },

{ "**name**": "Absolute Coordinates",
"**condition**": "No valid highlight available",
"**constraint**": "0-1000 scale (1000 = max dimension)",
"**format**": {"position": ["<x>", "<y>"]}
}

Figure 11: Visual Processing and Action Selection Prompt Template.

**Workflow Exception Handling Template:** Protocols for interrupting scenarios

"scenarios": [
{ "type": "Unexpected Pop-up",
"priority": 1,
"response": "Close before continuing main task",
"position_strategy": ["Index", "Relative coordinates > top-right corner"] },

{ "type": "Black Screen",
"response": "Wait 10s → Re-detect → Continue",
"timeout": "10000ms minimum" },

{ "type": "App Termination",
"method": "Swipe from center to screen edge",
"coordinate_spec": "start: [x_center, y_center], end: [x_edge≤1000, y_edge≤1000]" }
]

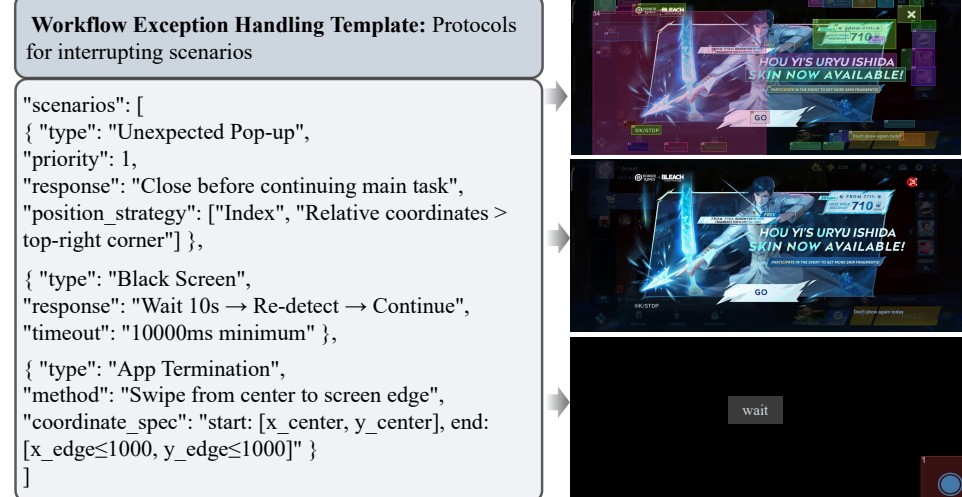

Figure 12: Exception Handling Prompt Template. Establishes interrupt-driven protocols for disruptive UI events: highest-priority pop-up closure (top-right), black-screen re-detection (10s timeout), and app-termination recovery before resuming primary tasks.

### E.3 ACTION SELECTION PROTOCOL

As depicted in Figure 11, the protocol formalizes a hierarchical targeting methodology for GUI interactions, prioritizing: (1) direct highlight indices when element-box alignment is exact; (2) relative coordinates (0.0-1.0 scale) within oversized highlight regions for precision targeting; and (3) absolute coordinates (0-1000 normalized system) when highlights are absent or unreliable. This tripartite selection strategy optimizes spatial accuracy while accommodating diverse interface topologies, with explicit constraints prohibiting coordinate values exceeding the 1000-unit boundary to maintain dimensional integrity.

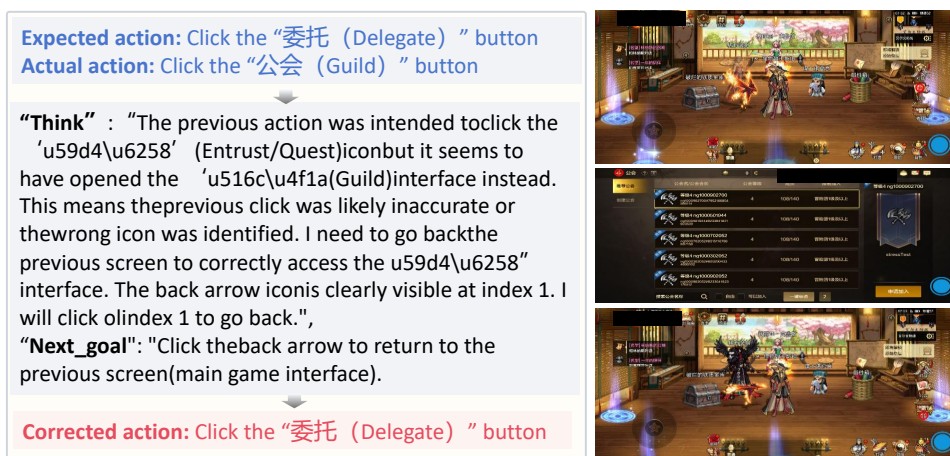

Figure 13: Error Recovery Example. Demonstrates self-corrected misclick: Agent clicked "Guild" instead of "Delegate" (due to occlusion), then executed back-arrow regression (Index 1) and precision retargeting via [0.5,0.8] coordinates to achieve the intended action.

### E.4 WORKFLOW EXCEPTION HANDLING

As illustrated in Figure 12, this template defines prioritized response protocols for disruptive interface events, establishing a scenario-based classification system: (1) unexpected pop-ups (highest priority, requiring immediate closure via top-right relative coordinates); (2) black screens (triggering 10-second re-detection cycles); and (3) background app termination (executed via edge-directed swipe vectors). The framework implements interrupt-driven workflow management, where exception resolution systematically precedes primary task progression to maintain environmental control stability.

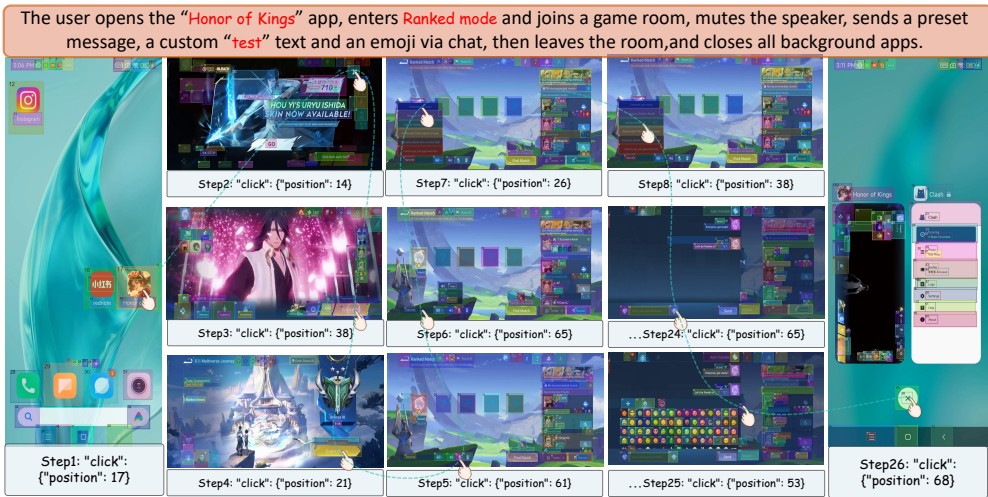

Figure 14: Game Scenario Case Visualization.

## F QUALITATIVE ANALYSIS

### F.1 ERROR CORRECTION VISUALIZATION

As illustrated in Figure 13, this sequence captures a critical error-recovery episode in our LongHorizonUI automation framework: The agent erroneously selected the adjacent "Guild" button instead of the target "Delegate" function, triggering an unintended guild management interface. Diagnostic self-assessment attributed this failure to positional deviation and visual occlusion interference

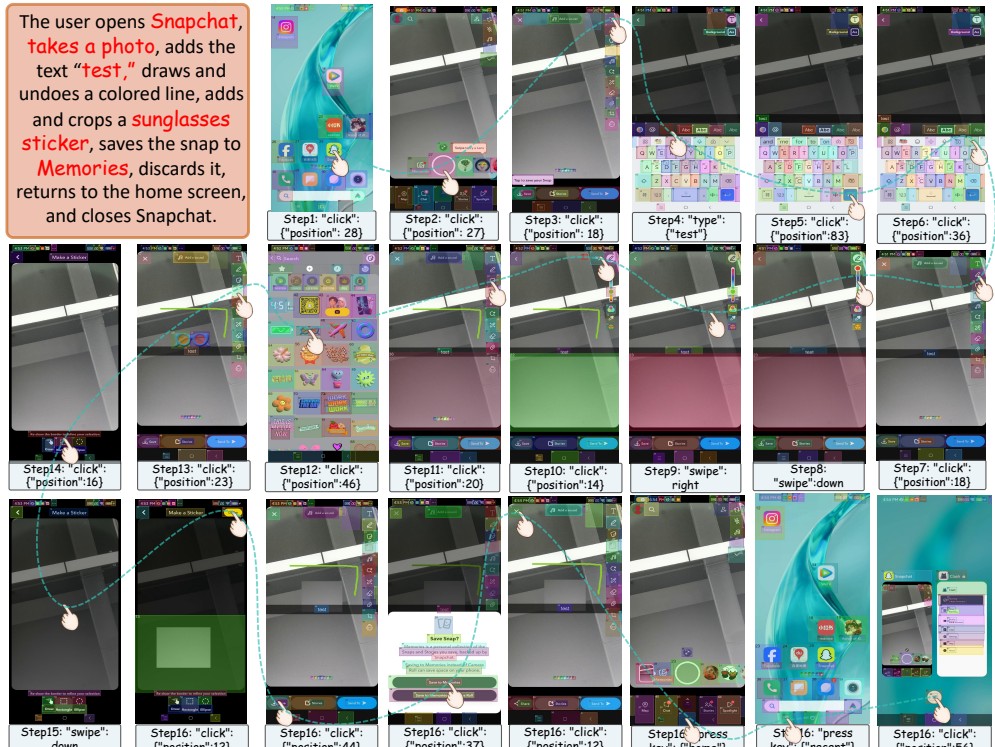

Figure 15: General Scenario Case Visualization.

within the GUI layout. To contain error propagation, the recovery protocol first activated a roll-back mechanism by clicking the back arrow (Index 1) to restore the baseline interface, followed by a precision-targeted secondary click using relative coordinates [N, 0.5, 0.8] within the Delegate button's highlight region, successfully rectifying the initial localization inaccuracy. This case demonstrates LongHorizonUI's operational efficacy and robustness in handling real-world automation exceptions.

### F.2 CASE VISUALIZATION

To demonstrate LongHorizonUI's advantage in long-horizon reasoning, we visualize its task execution trajectories in both general scenarios (Figure 14)and gaming environments (Figure 15). In universal settings, the architecture exhibits strong task generalization via its compensatory action executor, which dynamically adjusts interaction pathways when encountering heterogeneous UI elements (e.g., switching between gesture controls and traditional input fields) while maintaining task coherence. The deep-reflective decider further ensures minimal end-to-end error propagation by verifying stepwise contextual consistency, effectively mitigating cascading failures common in baselines. Within gaming scenarios, the agent leverages enhanced perceptual signals and compensatory action strategies to traverse nested menus and execute multi-step operations under real-time constraints, even during interface mutations.

## G ADDITIONAL DISCUSSIONS

The pursuit of robust long-horizon GUI agents necessitates addressing two critical challenges: adaptive long-horizon modeling and dynamic interrupt handling (e.g., pop-ups). For extended task sequences, future work could integrate reinforcement learning with hierarchical state representations to compress historical trajectories into abstract milestones, mitigating error accumulation while preserving contextual coherence. Insights from visual scene understanding—including semantic segmentation (Tian et al., 2022a; 2023; Lai et al., 2021; Tian et al., 2022b), few-shot segmentation (Tian et al., 2020; Peng et al., 2023; Luo et al., 2023; Ning et al., 2023), long-tailed and contrastive

representation learning (Cui et al., 2022; 2023), 3D scene understanding (Peng et al., 2024b; Wang et al., 2024a; Jiang et al., 2021; Wu et al., 2024a; Zhang et al., 2025a), text detection (Tian et al., 2019), and 3D content generation (Huang et al., 2025b)—offer promising directions for enriching the spatial and semantic perception capabilities of GUI agents. For dynamic interrupts (e.g., pop-ups), a predictive-reactive hybrid mechanism is essential: real-time environmental monitoring detects anomalies, triggering tiered fallbacks such as emergency rollbacks, LLM-guided diagnostics.

