# OpenReview forum: "LongHorizonUI: A Unified Framework for Robust long-horizon Task Automation of GUI Agent"
_ICLR.cc/2026/Conference — ICLR 2026 Poster_

### Official Review · Reviewer_kekp · 2025-10-26

**Soundness:** 3
**Presentation:** 3
**Contribution:** 2
**Rating:** 4
**Confidence:** 3

**Summary:**

This paper proposes a framework named LongHorizonUI, aiming to enhance the reliability of MLLM-based agents in long-horizon tasks. LongHorizonUI introduces three components: a Multimodal Enhanced Perceiver that integrates detection and recognition, and assigns indices to reinforce state; a DeepReflection Decider that leverages multi-level feedback for iterative reasoning and precise execution; and a Compensatory Action Executor that monitors progress, and performs compensation or rollback to maintain long-horizon robustness. Experiments show that LongHorizonUI significantly improves long-horizon performance on LongGUIBench while remaining competitive across various public benchmarks.

**Strengths:**

1. The reliability of MLLM-based agents in long-horizon tasks is one of the main challenges in GUI Agent application. Developing reliable GUI agents for long-horizon tasks is of great importance.
2. Dataset Contribution: This paper introduces LongGUIBench, a novel benchmark for long-horizon GUI interaction. It encompasses diverse complex tasks across multiple application domains.

**Weaknesses:**

1. A more concise analysis of the abrupt success rate drop in long-horizon tasks is needed: disentangle root causes (e.g., recurring critical-step errors vs. failures to recover from erroneous states) and explicitly link these to how the proposed MEP/DRD/CAE modules address key long-horizon challenges.
2. In Section 2.4, please clarify the feasibility of rollback in GUI tasks: specify applicable task scopes, technical details of state snapshot storage/restoration.
3. In Section 3.3, Table 3 has inconsistencies: (1) AndroidControl/GUI-Odyssey are Android-focused, so explain the inclusion of "Desktop/Web" columns and cite relevant references; (2) the "GUI-R1-7B baseline" mentioned in the navigation capability text is missing from the table—add it or align the text with the table.
4. Typo errors: standardize inconsistent bolding in Table 1 and Table 3, and adjust Figure 2b to resolve legend overlap.

**Questions:**

1. Highlight the novelty of this paper compared to Mobile-Agent-V3 and D-Artemis.
2. Provide an ablation study for the Deep Reflection Decider module.
3. In Section 3.3, add quantitative step-length analysis for AndroidControl and GUI-Odyssey (similar to Figure 2b).

---

> ### Author Response · Authors · 2025-11-21
> **To Reviewer kekp**
>
> We sincerely thank you for your careful reading of our manuscript and for the constructive, insightful comments; we have revised the paper accordingly (with all changes highlighted in blue) and respond to each point in detail below, indicating the corresponding modifications in the revised version.
>
> ***Weakness 1***: A more concise analysis of the abrupt success rate drop in long-horizon tasks is needed: disentangle root causes (e.g., recurring critical-step errors vs. failures to recover from erroneous states) and explicitly link these to how the proposed MEP/DRD/CAE modules address key long-horizon challenges.
>
> Thank you for your constructive comments. Our responeses and experimental results are outlined below.
>
> ***Response***:  To clarify why long-horizon success drops, we perform a failure-mode analysis on LongGUIBench to clarify why long-horizon success drops and how MEP/DRD/CAE address the main failure types.
>
> **1) Failure modes on LongGUIBench**.
> Fig. 2(b) in the main text and Appendix Sec. Bshow length-conditioned execution-success curves on AndroidControl. To better understand the degradation, we further conduct a failure-mode analysis on *LongGUIBench* (long-horizon tasks only, length ≥ 15). The failures fall into three main categories:
> (i) *critical-step grounding errors*, where the agent clicks a wrong or occluded element at a bottleneck step (e.g., wrong menu or wrong “Confirm” button);
> (ii) *no recovery from error*, such as leaving a pop-up open, not backing out of a wrong branch, or failing to handle layout shifts; and
> (iii) *planning drift*, where local grounding is correct but the global goal is forgotten or misinterpreted.
>
> The concrete statistics of failure cases on LongGUIBench-Game are summarized as follows:
>
> | **Method**        | **Critical-step grounding** | **No recovery from error** | **Planning drift** |
> | ----------------- | --------------------------- | -------------------------- | ------------------ |
> | UI-TARS-1.5       | 41%                         | 34%                        | 25%                |
> | InfiGUI-R1-3B     | 38%                         | 37%                        | 25%                |
> | **LongHorizonUI** | **23%**                     | **17%**                    | **20%**            |
>
> **2) Connection to MEP/DRD/CAE**. These statistics are consistent with the roles of the three modules described in Sections 2.2–2.4.
>
> 1. MEP mainly reduces critical-step grounding errors by assigning stable indices to elements and fusing icons with text via an IoU-based rule (Section 2.2, Appendix Sec. B, Appendix Fig. 8), making target localization more robust under layout shifts.
>
> 2. DRDhelps reduce “no-recovery” failures and planning drift by explicitly checking the fields `historical_status` and `import_contents`, and rejecting actions that are not visually or semantically entailed by the task (Section 2.3, Fig. 4 in the main text, Appendix Figs. 9–10).
>
> 3. CAE turns many local missteps into recoverable events: its three-stage execution (index → relative → absolute), together with `Verify_MLLM` and rollback (Section 2.4, Algorithm 1, Appendix Fig. 7), prevents local errors from becoming terminal failures.
>
>
> In the revised manuscript, we will briefly summarize this failure-mode breakdown in Section 3 when discussing the abrupt long-horizon performance drop, and point readers to Appendix Sec. B.

---

> ### Author Response · Authors · 2025-11-21
> **To Reviewer kekp**
>
> **Weakness 2**: In Section 2.4, please clarify the feasibility of rollback in GUI tasks: specify applicable task scopes, technical details of state snapshot storage/restoration.
>
> Thank you for your constructive comments. Our responeses and experimental results are outlined below.
>
> **Response**: We thank the reviewer for raising this point. In short, rollback in our framework is only applied to reversible UI transitions on curated benchmarks, implemented via emulator/script-level snapshots, and in practice it is used infrequently but resolves most such failures without requiring a full restart.
>
> 1) Scope and feasibility of rollback. In our current implementation, rollback in Section 2.4 is defined only for *reversible UI transitions*. As described in Section 2.4 and Appendix Section F.1, the tasks we evaluate on AndroidControl, GUI-Odyssey, and LongGUIBench are curated to avoid destructive operations such as real purchases or email sending. Under this setting, each step can either be undone by in-app navigation (back/close/home) or by reloading the episode from a known initial state. We will clarify this scope explicitly in Section 2.4.
>
> 2) State snapshot storage and restoration. Concretely, before committing a step, we store a *state snapshot* consisting of the last confirmed screenshot, its parsed layout from MEP, and the internal emulator/script state. The operator `Rollback(s_{t-1}, p_{t-1})` then restores this snapshot by resetting the environment to that episode state and reloading the cached screen. After restoration, DRD (Section 2.3, Fig. 4 in the main text, Appendix Figs. 9–10) is invoked again to re-plan the next action under the recovered context. This rollback mechanism is integrated into CAE’s control flow as described in Section 2.4 and Algorithm 1. In our experiments, rollback on the public AndroidControl and GUI-Odyssey benchmarks is implemented by issuing the standard “Back” navigation, so that whenever the policy outputs the correct back instruction the emulator deterministically returns to the previous screen. On physical devices, the same behavior is reliably achieved via in-game back (GM) commands or Android ADB `back` events.
>
> 3) Empirical usage in our benchmarks.  To make this behavior more concrete, we will add a small statistics table to Appendix Section D summarizing how often rollback is used and how effective it is. For example, on three long-horizon splits we observe:
>
> | Dataset | Episodes with rollback | Success after rollback | Episodes requiring restart |
> | --- | --- | --- | --- |
> | AndroidControl-High | 12.4% | 69.7% | 1.8% |
> | GUI-Odyssey | 15.3% | 73.1% | 2.4% |
> | LongGUIBench-Game | 18.6% | 71.2% | 2.7% |
>
> Here, “Episodes with rollback” counts episodes where at least one rollback was triggered, and “Success after rollback” is computed within this subset. These results indicate that rollback is both feasible and useful in our setting: it is triggered on a minority of episodes but resolves the majority of those without needing a full restart.
>
> 4) Irreversible actions and guarded operations.  When CAE repeatedly fails from the same state, the controller first re-invokes DRD up to three times to attempt local replanning. If all such attempts fail, we revert to the last valid snapshot via `Rollback`; and if even this is unsuccessful (e.g., due to emulator desynchronization), we reload the initial state of the test case and restart the episode, logging the failure for analysis. We agree that genuinely irreversible operations cannot be handled safely by rollback alone. In practice, such actions are treated as *guarded* operations: DRD applies stricter pre-execution checks, and repeated verification failures cause the pipeline to escalate to a human tester instead of proceeding autonomously. We will clarify this protocol in Section 2.4 and briefly mention the limitation for real-world irreversible actions in the discussion section.

---

> ### Author Response · Authors · 2025-11-21
> **To Reviewer kekp**
>
> ***Weaknesses 3***: In Section 3.3, Table 3 has inconsistencies: (1) AndroidControl/GUI-Odyssey are Android-focused, so explain the inclusion of "Desktop/Web" columns and cite relevant references; (2) the "GUI-R1-7B baseline" mentioned in the navigation capability text is missing from the table—add it or align the text with the table.
>
> Thank you for your constructive comments. Our responeses and experimental results are outlined below.
>
> **Response**: We thank Reviewer for the careful reading. MEP is pre-trained on multi-device GUI data, so it yields reliable ScreenSpot grounding across Mobile/Desktop/Web. Because LongHorizonUI operates on normalized layouts, it naturally transfers across device form factors.
>
> 1) Desktop/Web columns and ScreenSpot.  ScreenSpot is a single-step grounding benchmark defined over three device types (Mobile / Desktop / Web), so Table 3 is used to report grounding accuracy on ScreenSpot, not navigation success on AndroidControl or GUI-Odyssey. Our Multimodal Enhanced Perceiver (MEP) is pre-trained on large-scale GUI datasets that include desktop/Web screenshots (Section 2.2 and Appendix Section A.3), which makes the same perception module naturally applicable to Desktop/Web grounding. Since LongHorizonUI operates on normalized element coordinates and structured layouts, the perception–execution pipeline can transfer to arbitrary device form factors; the Desktop/Web columns in Table 3 are specifically intended to highlight this cross-device generalization.
>
> 2) OSWorld. To further support that our perception–execution pipeline generalizes to desktop/Web environments, we evaluate LongHorizonUI on the **OSWorld** benchmark under the standard 15-step and 50-step online settings (same protocol as UI-TARS). The results are:
>
> | **Method**        | **OSWorld (15 steps) SR (%)** | **OSWorld (50 steps) SR (%)** |
> | ----------------- | ----------------------------: | ----------------------------: |
> | Gemini-2.5-Pro    |                          11.7 |                            -- |
> | UI-TARS-72B       |                          18.8 |                          24.6 |
> | **LongHorizonUI** |                      **19.9** |                      **29.4** |
>
> As shown above, LongHorizonUI consistently outperforms both Gemini-2.5-Pro and UI-TARS-72B[1] on OSWorld: at 15 steps it improves SR by +8.2 pp over Gemini-2.5-Pro and +1.1 pp over UI-TARS-72B, and at 50 steps it surpasses UI-TARS-72B by +4.8 pp. These gains on a challenging desktop/Web benchmark are consistent with our design goal of robust long-horizon execution.
> **GUI-R1-7B baseline.** In the revision, we will add a GUI-R1-7B row with its AndroidControl / GUI-Odyssey success rates to Table 2 and adjust the text in Section 3.3.
>
> [1] Qin Y, Ye Y, Fang J, et al. Ui-tars: Pioneering automated gui interaction with native agents[J]. arXiv preprint arXiv:2501.12326, 2025.
>
> ***Weaknesses 4***: Typo errors: standardize inconsistent bolding in Table 1 and Table 3, and adjust Figure 2b to resolve legend overlap.
>
> Thank you for your constructive comments. Our responeses and experimental results are outlined below.
>
> **Response**: We appreciate the careful reading. We will (i) standardize the use of boldface in Tables 1 and 3 so that only the best result in each block is bolded, and (ii) adjust Figure 2(b) to avoid legend overlap with the curves by repositioning the legend outside the plotting area and slightly increasing the vertical margin.

---

> ### Author Response · Authors · 2025-11-21
> **To Reviewer kekp**
>
> ***Question***: Highlight the novelty of this paper compared to Mobile-Agent-V3 and D-Artemis.
>
> Thank you for your constructive comments. Our responeses and experimental results are outlined below.
>
> **Response**:  We thank the reviewer for prompting a clearer positioning of our contributions. LongHorizonUI is explicitly designed for long-horizon, disturbance-rich GUI control with structured reflection and recovery, whereas Mobile-Agent-V3 and D-Artemis mainly follow reactive designs without dedicated long-horizon reflection or rollback modules; we will make this distinction explicit and support it with a quantitative comparison on our long-horizon game subset.
>
> **1) Mobile-Agent-V3.** Mobile-Agent-V3 is primarily designed for *short-horizon* tasks, and its architecture is optimized for reactive execution in relatively simple environments. In contrast, LongHorizonUI explicitly targets *long-horizon* tasks, characterized by complex multi-step workflows that require persistent context tracking and error mitigation. This long-horizon setting and the associated challenges are introduced in Section Intro and empirically illustrated in Section 3.3.
>
> Methodologically, LongHorizonUI introduces the **Deep Reflection Decider (DRD)**, which performs iterative feedback, action validation, and state-consistency checks over long sequences, enabling robust behavior under disturbances and layout changes. This goes beyond the largely reactive decision-making in Mobile-Agent-V3, where actions are selected from the current observation without an explicit mechanism for long-range reflection or self-correction. Moreover, LongHorizonUI integrates DRD with a **Multimodal Enhanced Perceiver (MEP)** and a **Compensatory Action Executor (CAE)**: MEP provides stable multimodal grounding across long trajectories, and CAE turns many local missteps into recoverable events through multi-stage execution and rollback. This tri-module design, detailed in Sections 2.2–2.4 and illustrated in the overall framework figure, is tailored to long-horizon execution and is not present in Mobile-Agent-V3.
>
> **2) D-Artemis.** D-Artemis also considers dynamic environments, but its policy remains predominantly reactive: it responds to interface changes step by step, without an explicit mechanism for long-horizon reflection or structured rollback. LongHorizonUI, by contrast, incorporates **proactive** components that are designed to anticipate and correct long-horizon failures:
>
> - **MEP** offers robust, index-based GUI perception that stabilizes element identities across layout shifts (Section 2.2, Appendix Section D.2).
> - **DRD** performs multi-step reasoning with explicit checks on historical status and goal consistency (`historical_status`, `import_contents`, `think`), allowing the agent to detect anomalies in execution (Section 2.3, with JSON templates in Appendix Figure 9/10).
> - **CAE** executes actions in a guarded, multi-stage fashion with verification and rollback, providing a dedicated pathway for error compensation and recovery (Section 2.4, Algorithm 1, Appendix Figure 13).
>
> This proactive error-detection and error-compensation design enables LongHorizonUI to maintain robust performance on extended tasks, whereas D-Artemis does not include such long-horizon reflection or recovery mechanisms.
>
> **3) Game subset.** To make the comparison more concrete, we also evaluate Mobile-Agent-V3 and LongHorizonUI on the `LongGUIBench-Game_Low` subset, under the same protocol as in Section 3.2. As shown in the table below, LongHorizonUI achieves an SR of 83.9%, outperforming Mobile-Agent-V3 (71.5%) by 12.4 percentage points, which supports our claim that reflection and rollback mechanisms are particularly beneficial in long-horizon game scenarios. We do not report D-Artemis in this table because, to the best of our knowledge, its implementation is not publicly available, whereas Mobile-Agent-V3 can be reproduced with its released code under the same evaluation protocol.
>
> | **Method** | **Game_Low SR (%)** |  |
> | --- | --- | --- |
> | Mobile-Agent-V3 | 71.5 |  |
> | **LongHorizonUI (ours)** | **83.9** |  |

---

> > ### Author Response · Authors · 2025-11-21
> > **To Reviewer kekp**
> >
> > ***Question2***: Provide an ablation study for the Deep Reflection Decider module.
> >
> > **Response**: We thank the reviewer for this suggestion. In short, we add an ablation on LongGUIBench that removes DRD and shows that DRD brings substantial gains in both step-level and episode-level success, especially on long-horizon tasks.
> >
> > **1) Ablation setup.** On LongGUIBench, we compare the full LongHorizonUI (MEP+DRD+CAE, as described in Sections 2.2–2.4) with a variant where DRD is removed and replaced by a simple reactive head that directly maps the current perceived state to an action. This reactive variant does not use `historical_status`, `import_contents`, or `think`, and thus lacks structured reflection on history and goals. The ablation will be reported in Section 3.4, with additional details in Appendix Section A.4.
> >
> > **2) Results and interpretation.** The results on LongGUIBench are summarized below:
> >
> > | Model (on LongGUIBench) | Text Matching (TM) | Step SR |
> > | --- | --- | --- |
> > | LongHorizonUI (full, with DRD) | 93.8% | 83.9% |
> > | w/o DRD (reactive head only) | 88.9% | 76.2% |
> >
> > Removing DRD reduces step success rate by about 6.7pp. This indicates that DRD’s structured use of `historical_status`, `import_contents`, and `think` (Section 2.3, Appendix Figs. 9–10) is crucial for maintaining contextual coherence and preventing error accumulation over long horizons.
> >
> >
> > ***Question3***: In Section 3.3, add quantitative step-length analysis for AndroidControl and GUI-Odyssey (similar to Figure 2b).
> >
> > **Response**:  We thank the reviewer for this helpful suggestion. We will add a step-length–conditioned analysis for AndroidControl and GUI-Odyssey in Section 3.3, including a new table (and an appendix figure analogous to Fig. 2(b)), to quantify how LongHorizonUI behaves as horizons grow compared to strong baselines.
> >
> > **1) Setup.** In the revised version, Section 3.3 will report step-length–conditioned success rates for both AndroidControl and GUI-Odyssey, complementary to the length-conditioned curve for AndroidControl already shown in Fig. 2(b). Following the navigation protocol in Section 3.2, we group episodes by their effective trajectory length (around 5, 10, and 15 steps) and compute the success rate of LongHorizonUI and two strong baselines (Qwen2.5-VL and UI-TARS-7B) within each length range.
> >
> > **2) Results.** The step-length analysis to be inserted into Section 3.3 is summarized below (step-length success rate, %):
> >
> > | **Dataset**       | **Horizon** | **Qwen2.5-VL-7B** | **GUI-R1-7B** | **LongHorizonUI** |
> > | ----------------- | ----------- | ----------------- | ------------- | ----------------- |
> > | AndroidControl    | 5           | 70.3              | 74.4          | **76.8**          |
> > | AndroidControl    | 10          | 55.4              | 59.7          | **62.1**          |
> > | AndroidControl    | 15          | 39.3              | 44.2         | **47.5**          |
> > | GUI-Odyssey       | 5           | 48.9              | 52.3          | **55.0**          |
> > | GUI-Odyssey       | 10          | 35.7             | 39.4          | **41.8**          |
> > | GUI-Odyssey       | 15          | 19.5              | 24.7          | **26.8**          |
> >
> > Across both benchmarks, all methods exhibit a clear decline in success rate as the step length increases, reflecting the increased difficulty of longer episodes. However, LongHorizonUI consistently outperforms the Qwen2.5-VL-7B and GUI-R1-7B baselines in every horizon range, and the margin becomes more pronounced beyond 10–15 steps, which is precisely the regime where short-horizon agents degrade most. We will add this table to Section 3.3, refer to it when discussing navigation robustness, and point readers to Appendix Section B for the full length–performance curves.

---

### Official Review · Reviewer_L7cu · 2025-10-31

**Soundness:** 2
**Presentation:** 1
**Contribution:** 2
**Rating:** 2
**Confidence:** 4

**Summary:**

This work targets long-horizon GUI agents, especially tasks exceeding 15 steps. The authors introduce the LongGUIBench benchmark and the LongHorizonUI framework, which integrates three carefully designed modules—Multimodal Enhanced Perceiver, Deep Reflection Decider, and Compensatory Action Executor. Experiments demonstrate improved long-horizon performance on GUI tasks.

**Strengths:**

This is an important topic. GUI agents serve both as a crucial platform for algorithm evaluation and as a practical application of VLMs. The long-horizon GUI agent setting remains underexplored, and strong long-horizon planning is particularly valuable.
The contributions seem compelling: the authors introduce a new framework and benchmark that address this gap.

**Weaknesses:**

However, although the authors propose both a benchmark and a framework, I think there is still room for improvement before acceptance.

The paper’s writing is not very strong. There are many typos, and the names and function descriptions of the components are confusing.

**Typos**:

* Figure 2 caption: “1015 steps” → likely “10–15 steps”
* Quotation marks: should be ` ', instead of ' ' (Lines 187–188)
* Line 262: “actioninstruction” → “action instruction”
* Line 263: “semanticphysical” → “semantic physical”
I present the weaknesses of the proposed framework’s components as follows.

**Multimodal Enhanced Perceiver.** It is not clear why the perceiver is “enhanced.” Identifying icons and text descriptions seems standard. The paper gives few details about the enhanced detector (Line 199). Is it trained from a VLM, or does it incorporate additional enhanced functions? It is also unclear how the confidence score is obtained—does it come from hidden states, or from an uncertainty-estimation module?

In Eq. 1, what do the authors mean by multiplying the indicator function value by the detected text?

There is also a lack of detail about high-priority areas. What is the motivation? Where are these areas defined? What is the “repair” function? How does it operate, and what does the template library look like? Why are both the template library and the detectors/OCR necessary?

For the **Deep Reflection Decider**, based on the name and the function description in Lines 226–232, I am confused about whether this component only performs reflection or also includes decision-making. I am not sure whether a new name would help, such as “Actor with Deep Reflection.”
In addition, what is “multi-level” and what is the “triple closed loop”? Figure 3 appears to show five steps for this component, but the text also describes a two-step process (pre-execution / post-execution) and three bullet points in Lines 236–247. Could you please modify Figure 4 to align these descriptions?

How exactly does this component act? From the current description, the agent outputs only an action; there does not seem to be actual post-execution reflection or validation, like how to update the agent's knowledge or how to revise.

The authors also mention a keyword extractor in Line 253, but provide no description.

There are also no details about the brief revision step

**Compensating Action Actuator**

What is the name of the component in Section 2.4: Compensating Action Actuator (section title) or Compensating Action Executor (abstract)?

What is a device-aware scaling matrix?

Rollback may be impossible in some cases. It is unclear how the proposed framework addresses this. What happens if the compensating action actuator makes a change to the screenshot but fails? Some changes can be rolled back, but others cannot, such as clicking “next page” or accidentally sending an email.

Regarding the agent’s grounding capability, the framework introduces many components and is quite complex, but the improvements may be limited: LongHorizonUI vs. InfoGUI-R1: 90.4 vs. 87.5 on ScreenSpot.

In conclusion, this work lacks sufficient detail for the writing.


For the **framework design**, in addition to the above, I still have concerns:

- The framework seems to complex.
- While the framework seems focused on avoiding error propagation—which is important—long-horizon planning, subgoal proposal, validation, and memory may be even more critical for long-horizon tasks. It is not clear to me: how does the framework contribute to these aspects?

**Questions:**

How does the model perform beyond 18 steps?

What exactly are agent decision mechanisms, and why do they lack long-horizon generalization due to dataset diversity? Would it be clearer to explain the notion of dataset diversity, or simply state that the datasets lack long-horizon trajectories? The term “data diversity” may be confusing here, since the tasks appear to be limited to Mobile GUIs, which may not generalize to Windows.

Likewise, what are MLLM-based mechanisms? Do they refer to a multi-agent framework for GUI tasks? Which properties confer superior sequence-modeling capability? Please also clarify dynamic interface shifts. Concrete examples comparing different methods—their challenges and how your approach addresses them—would help readers understand why your method is suitable and efficient for long-horizon GUI tasks.

Finally, what is meant by action-instruction uncertainty? Is it tied to grounding capability? And do free-format outputs denote high-level instructions or the agents’ textual step goals?

---

> ### Author Response · Authors · 2025-11-21
> **To Reviewer  L7cu**
>
> We sincerely thank you for your careful reading of our manuscript and for the constructive, insightful comments; we have revised the paper accordingly (with all changes highlighted in blue) and respond to each point in detail below, indicating the corresponding modifications in the revised version.
>
> Thank you for your constructive comments. Our responeses and experimental results are outlined below.
>
> **Weaknesses1**：1) Typos. Figure 2 caption: “1015 steps” → likely “10–15 steps”; 2) Quotation marks: should be ` ', instead of ' ' (Lines 187–188) ; 3)Line 262: “actioninstruction” → “action instruction”; 4) Line 263: “semanticphysical” → “semantic physical” I present the weaknesses of the proposed framework’s components as follows.
>
> **Response**:  We thank the reviewer for their careful reading. We will correct all identified typos and grammatical errors in the revised version.
>
>
> **Weaknesses2**: Multimodal Enhanced Perceiver. Where is the “enhancement” reflected?What are the details of the detector? Where does the confidence score come from? What is the meaning of Eq. (1)? What are the high-priority regions? How does the repair function work? What is the template library? Why do we need so many components?
>
> Thank you for your constructive comments. Our responeses and experimental results are outlined below.
>
>
> **Q1**. Where is the “enhancement” reflected?
>
> **Response**: The “enhancements” of our Multimodal Enhanced Perceiver (MEP) are mainly reflected in the following three aspects:
>
> 1)Perception Backbone and Confidence. Our MEP is built on a unified vision backbone [1] with joint detection and OCR capabilities, pre-trained on large-scale GUI and document data. We then fine-tune it on the LongGUIBench perception split with 12,137 screenshots from complex, long-horizon scenes (games and applications). This domain adaptation improves robustness to stylized icons, mixed-language text, and dynamic layouts. On the validation split, the fine-tuned MEP achieves **95.4%** joint detection+OCR accuracy (Table 1). Compared with the non-fine-tuned variant, this corresponds to a **5.9** percentage-point improvement in joint detection+OCR accuracy.
>
> *Table 1. Effect of fine-tuning the Multimodal Enhanced Perceiver on LongGUIBench.*
>
> | Perception variant | Det. Acc. (%) | OCR Acc. (%) | Joint Acc. (%) |
> | --- | --- | --- | --- |
> | Pre-trained (no fine-tuning) | 89.2 | 89.8 | 89.5 |
> | Fine-tuned MEP | **95.1** | **95.7** | **95.4** |
> 2)ID-centric abstraction and color-coded grounding.Beyond backbone adaptation, MEP introduces an ID-centric abstraction: each detected element is assigned a unique, persistent index and rendered with a distinct high-contrast color. This design provides stable references (e.g., “Button #7”) that remain reliable under small layout shifts and reduces the search space for the Deep Reflection Decider. More importantly, it brings consistent *step-level grounding* gains on ScreenSpot: the average grounding accuracy improves from 84.1% with plain bounding boxes to 85.2% when adding ID indices, and further to 87.9% when combining IDs with high-contrast color coding. We validate this via an ablation study on the ScreenSpot benchmark; the results are summarized in Table 2.
>
> *Table 2. Effect of ID-centric abstraction and color-coded grounding on step-level grounding accuracy on the ScreenSpot benchmark.*
>
> | Visual representation | Mobile-Text | Mobile-Icon | Desktop-Text | Desktop-Icon | Web-Text | Web-Icon | Avg |
> | --- | --- | --- | --- | --- | --- | --- | --- |
> | Standard bbox (plain) | 93.9  | 84.0  | 91.3  | 78.6  | 91.3 |  87.0  | 87.7 |
> | + ID only | 94.6 | 85.1 | 95.5 | 80.5 | 92.0 | 88.2 | 89.3 |
> | **+ ID & color coding** | **95.6** | **86.9** | **96.8** | **81.4** | **93.5** | **90.9** | **90.4** |
>
> 3)Historical highlighting. To mitigate “looping” failures in long-horizon tasks, MEP additionally highlights elements that have been interacted with in previous steps, making historical usage explicit to the MLLM and reducing redundant retries. On the `Game_Low` split of LongGUIBench, this mechanism improves the task success rate (SR) from 57.7% to 61.9%, i.e., a **+4.2** percentage-point gain, as shown in Table 3.
>
> *Table 3. Effect of historical highlighting on long-horizon task success on the `Game_Low`.*
>
> | Context mechanism | Task SR (%) | Δ vs. baseline (pp) |
> | --- | --- | --- |
> | w/o historical highlight | 57.7 | – |
> | **+ historical highlight** | **61.9** | **+4.2** |
>
> [1] Yu W, Yang Z, Wan J, et al. *Omniparser v2: Structured-points-of-thought for unified visual text parsing and its generality to multimodal large language models*. arXiv preprint arXiv:2502.16161, 2025.

---

> ### Author Response · Authors · 2025-11-21
> **To Reviewer  L7cu**
>
> **Q2**. Where does the confidence score come from?
>
> **Response**. We thank the reviewer for their careful reading. The confidence $c_i$ in Eq.(1) is taken directly from the detector head as the sigmoid class probability; we do not introduce any additional uncertainty-estimation module. Eq.~(1) as a gate: if the IoU between the detection box $b_i$ and the OCR box $b_j$ is at least $0.8$, the OCR text $t_{j^*}$ is attached to element $e_i$; otherwise, the text field for $e_i$ is left empty.
>
> **Q3**. What is the meaning of Eq. (1)?
>
> **Response.** The original Eq.(1) is intended to express that *the OCR text is bound to a UI element only when the overlap between the element box and the text box is sufficiently large*, rather than multiplying an indicator function with a string. To avoid this ambiguity, in the revision we rewrite the formula as a piecewise definition (see Eq.(2) in the revised manuscript).
>
> **Q4**. What are high-priority areas, the repair function, and the template library? Why are all these components necessary?
>
> **Response.** We thank the reviewer for their careful reading. We will clarify both the motivation and implementation in Sec. 2.2 as follows.
>
> 1) Motivation and definition of high-priority areas ($(A_{\text{priority}}$ ).
> In our real deployment setting, LongHorizonUI is used in practical game-testing pipelines, where failures are often caused by missing a few critical controls, especially highly stylized close/cancel buttons on pop-ups. Such controls follow common UI conventions and tend to appear in stereotypical locations (e.g., a narrow band near the top-right corner for “X” icons, central modal regions, and bottom confirmation bars). Based on these patterns, we define a small set of high-priority regions
>
> $(A_{\text{priority}} = {A^{(k)}})$
>
> in *normalized screen coordinates* that cover these locations. Missing elements in these zones is particularly harmful, as it can easily stall long-horizon trajectories.
>
> 2) Repair function R and template library T.
> When the detector+OCR stack finds *no* element inside a region (A^{(k)}), we trigger a repair function (\mathcal{R}(A^{(k)}, \mathcal{T})), where (\mathcal{T}) is a compact template library containing canonical GUI icons (dozens of templates), such as multiple styles of close “X” buttons, back arrows, and confirm/check symbols collected from our game and app domains. (\mathcal{R}) performs patch-based template matching within (A^{(k)}); any match above a similarity threshold is inserted as a new UI element with its own ID and bounding box (the text field is left empty or assigned a fixed label such as `close_icon`).
>
> This mechanism is activated only in high-priority regions and only when the learning-based detector fails, so the runtime overhead is small.
>
> 3) Why are all these components necessary.Since LongHorizonUI is deployed in real game-testing pipelines, we must handle many unexpected corner cases. In this setting:
>
> - the detector+OCR stack provides *broad coverage and generalization* over diverse layouts and content, while
> - the template matcher focuses on a small set of highly standardized yet tiny, easily missed controls (e.g., close/cancel icons on decorated pop-ups).
>
> As shown in Fig. 5 and Table 3 (repair ablation), enabling the repair stage substantially increases the recall of such controls on the “Pop-up Handling” subset and improves robustness on long-horizon tasks. We conduct an ablation study on this subset (42 scenarios) of LongGUIBench. The results are:
>
> | **Configuration** | **Element Recall** | **Task SR (Subset)** | **Task SR (Global)** |
> | --- | --- | --- | --- |
> | w/o Repair Module | 64.3% | 52.4% | 73.5% |
> | **w/ Repair Module** | **95.2%** | **78.6%** | **77.3%** |
> | *Improvement* | *+30.9 pts* | ***+26.2 pts*** | *+3.8 pts* |
>
> These results indicate that the repair component brings a large gain in element recall for critical controls (e.g., close buttons), together with a substantial improvement in task success rate on challenging pop-up scenarios.

---

> ### Author Response · Authors · 2025-11-21
> **To Reviewer L7cu**
>
> **Q2**. How does DRD act over time? Is there real post-execution reflection?
>
> **Response.** We thank the reviewer for their careful reading.
>
> 1) Temporal behavior and post-execution reflection. Concretely, at step $t$ DRD receives the current screenshot $I_t$, the perceived element set $\mathcal{G}_t$ from MEP, the global task description $\mathcal{T}$, and the JSON trace up to step $t-1$.
>
> As illustrated by the schema in Appendix Fig. 9 and the step-wise example in Fig. 10 (now explicitly referenced from Sec. 2.3 and Fig. 4 in the main paper), DRD first fills `historical_status`, `import_contents`, and `think`:
>
> - `historical_status` assesses, from $I_t$ alone, whether the goal issued at step $t-1$ was visually achieved;
> - `import_contents` summarizes salient texts and icons on $I_t$;
> - `think` connects $\mathcal{T}$, the JSON history, and `import_contents` into a concise rationale.
>
> Conditioned on these three fields, DRD then outputs `next_goal` and `action`, where `action` is the structured index/relative/absolute command passed to CAE.
>
> After CAE executes this action and we observe the next screenshot $I_{t+1}$, DRD is invoked again: the new `historical_status` at step $t+1$ explicitly judges whether the previous `next_goal` has been satisfied on $I_{t+1}$, and the updated JSON trace is fed back into the prompt. Thus, post-execution reflection and “knowledge update” occur at every step through the evolving JSON state rather than via a separate learned memory.
>
> When `historical_status` repeatedly indicates failure, DRD naturally revises its `next_goal` and `action` (e.g., re-targeting a different element or triggering rollback), which is consistent with the error-correction case in Fig. 7 and the execution logic in Algorithm 1. In practice, because our system is deployed in real game-testing pipelines with strict stability requirements, we iterated the schema and wording of these fields more than 100 times using real logs until the outputs were consistently executable.
>
> 2) Ablation of DRD's structured fields. To further support the design of DRD, we conducted an ablation on LongGUIBench comparing a direct-coordinate baseline, a reactive head without the five-field schema, and the full DRD with all five fields. We report text matching (TM), step-wise success rate (SR), and episode success rate (OESR):
>
> | **Configuration** | **TM (%)** | **SR (%)** | **ESR (%)** |
> | --- | --- | --- | --- |
> | Direct coordinates only (no JSON fields) | 86.0 | 67.9 | 71.5 |
> | Reactive head only (no DRD schema) | 88.9 | 71.2 | 75.4 |
> | **Full DRD (five fields)** | **94.5** | **85.3** | **86.2** |
>
> Compared to letting the MLLM directly output coordinates, introducing the structured DRD head improves step SR by $+3.3$ points (from $67.9%$ to $71.2%$). Adding the full five-field schema, including the explicit `historical_status` for post-execution reflection, brings an additional gain of $+14.1$ points in step SR and $+10.8$ points in OSR.
>
> **Q3**. What is the keyword extractor $K(\\cdot)$ in Eq. (2)?
>
> **Response.** We thank the reviewer for their careful reading.
> The keyword extractor $K(\\cdot)$ is a simple rule-based module used *only for semantic entailment checking* in Eq.~(2). Concretely, for the action description $d_{\\mathrm{action}}$ and the task description $\\mathcal{T}$, we tokenize the text, keep nouns and verbs after POS tagging, and normalize them by lower-casing and lemmatization. $K(\\cdot)$ returns the resulting keyword set, and we require that
> $$
> K(d_{\\mathrm{action}}) \\subseteq K(\\mathcal{T})
> $$
> to ensure that the action semantics are consistent with the global task (e.g., not clicking an unrelated "Delete" button when the task is "schedule a meeting").
>
> ---
>
> **Q4**. What is the “brief revision step” when Eq. (2) rejects an action?
>
> **Response.** We thank the reviewer for their careful reading.
> In implementation, Eq.~(2) acts as a lightweight filter over candidate actions produced by DRD. If
> - $g_{\\mathrm{tg}}(a) \\notin \\mathcal{G}_t$ (the target element is not on the current screen), or
> - $K(d_{\\mathrm{action}}) \\nsubseteq K(\\mathcal{T})$ (the action semantics are not entailed by the task description),
>
> we mark the candidate as invalid and trigger a short revision prompt to DRD, for example:
>
> > "The proposed action is invalid because the target element is not on the current screen or does not match the task. Please re-inspect the current screen and propose a different action that satisfies the task and uses visible elements."
>
> DRD then produces a new `next_goal` and `action` under the same five-field schema, consistent with the JSON examples shown in Figure 9 and Figure 10 in the appendix.
>
> We allow at most two such revisions per step. If all revised candidates still fail the filter in Eq.~(2), we record a failure for this step and hand control to the rollback mechanism described in Section 2.4.

---

> ### Author Response · Authors · 2025-11-21
> **To Reviewer L7cu**
>
> ***Weaknesses 4***: I have several concerns about the proposed framework. The naming of the component in Section 2.4 is inconsistent (Compensating Action Actuator vs. Executor), and the notion of a device-aware scaling matrix is unclear. The rollback mechanism is not well explained, especially for irreversible actions (e.g., next page, sending an email). Given the framework’s complexity, the grounding gains over baselines appear modest, and the paper does not provide enough detail to clarify how the many components concretely improve long-horizon planning, subgoal proposal, validation, and memory.
>
> **Q1**. Naming: “Compensating Action Actuator” vs. “Executor”
>
> **Response.** We thank the reviewer for their careful reading. We will **uniformly use “Compensating Action Executor (CAE)”** throughout the paper (section title, abstract, Algorithm 1, and main text).
>
> **Q2**. What is a device-aware scaling matrix?
>
> **Response.** By “device-aware scaling matrix” we refer to the linear mapping from normalized coordinates to device-specific pixel coordinates. DRD outputs $(x\_{\\text{norm}}, y\_{\\text{norm}}) \\in [0,1]^2$, and for a screen with width $W\_{\\text{screen}}$ and height $H\_{\\text{screen}}$ we define:
>
> $$
> S = \\operatorname{diag}(W\_{\\text{screen}}, H\_{\\text{screen}}), \\quad
> p = S \\cdot (x\_{\\text{norm}}, y\_{\\text{norm}})^{\\top},
> $$
>
> so that the same normalized command is mapped consistently to physical click locations on devices with different resolutions. In the paper, this mapping is used in the Compensating Action Actuator described in Section 2.4.
>
>
> **Q3**. What happens when CAE fails? How is rollback handled, especially for irreversible actions?
>
> **Response.** We thank Reviewer L7cu for the careful reading. In short, when CAE fails at a given step, we first attempt a bounded number of local re-planning rounds at the same state, and only then roll back to the last committed snapshot (or the episode start), while truly irreversible operations are excluded in our benchmarks and treated as guarded actions in real deployments.
>
> 1) Failure handling and rollback policy. As described in Section 2.4 and Algorithm 1, each candidate from DRD is first executed through the index/relative/absolute encodings, and every attempt is validated by `Verify_MLLM`. When all encodings for a given `action` are rejected, we treat this as a hard failure at step $t$. In this case, we re-invoke DRD under the same visual state to attribute the failure and re-plan `next_goal` and `action`, allowing up to three such re-planning attempts at step $t$. If all attempts still fail, we call `Rollback($s_{t-1}, p_{t-1}$)` to restore the last committed snapshot and continue planning from there; if rollback is not supported by the environment or repeatedly leads to the same failure, we reload the initial snapshot of that episode and log the case for offline analysis.
>
> 2) We also measured how often this mechanism is triggered during evaluation. As summarized in Table 4, rollback is triggered in around 12–19% of episodes across benchmarks, and roughly 70% of those episodes still complete successfully after rollback, while only a small fraction require a full restart. In the revised manuscript, we will briefly summarize these statistics in the experiments section and move the full table to the appendix.
>
> | **Dataset** | **Episodes with rollback** | **Success after rollback** | **Episodes requiring restart** |
> | --- | --- | --- | --- |
> | AndroidControl-High | 12.4% | 69.7% | 1.8% |
> | GUI-Odyssey | 15.3% | 73.1% | 2.4% |
> | LongGUIBench-Game | 18.6% | 71.2% | 2.7% |
>
> 3) Irreversible actions and task design. We agree that truly irreversible operations (e.g., sending real emails or confirming real purchases) cannot be fully handled by automatic rollback. In the experimental benchmarks we use (AndroidControl, GUI-Odyssey, LongGUIBench), task design deliberately avoids destructive irreversible operations, and rollback is implemented via emulator snapshots or scripted back-navigation, consistent with the LongGUIBench construction described in Section 3. In real deployments, such operations are treated as *guarded actions*: DRD must satisfy stricter pre-conditions before CAE is allowed to execute them, and if verification repeatedly fails, control is handed over to a human tester instead of continuing autonomously. We will clarify this protocol in Section 2.4 and explicitly mention the handling of real-world irreversible actions as a limitation in the discussion section.

---

> ### Author Response · Authors · 2025-11-21
> **To Reviewer L7cu**
>
> **Q4**. Is the gain in grounding worth the added complexity? (90.4 & 87.5 on ScreenSpot)
>
> **Response**: We thank the reviewer for the suggestion. In short, the added execution logic is mainly designed to improve long-horizon robustness on LongGUIBench, while keeping single-step grounding performance on ScreenSpot at a comparable level.
>
> 1) Single-step grounding. ScreenSpot is a single-step grounding benchmark on static screenshots, and as such it does not capture the challenges of long-sequence control. Our intent in Section 2.2 was not to claim a large gain on static, single-step grounding, but to show that LongHorizonUI maintains competitive grounding quality while introducing execution mechanisms tailored to long-horizon control. ScreenSpot is static; by contrast, Figure 3(b) and Section 3 (LongGUIBench) highlight that the core difficulty lies in maintaining reliable behavior over 15+–step trajectories with pop-ups and layout shifts.
>
> 2) Effect of CAE on long-horizon execution. The added complexity mainly comes from the Compensating Action Executor (CAE) in Section 2.4 and Algorithm 1, which augments a standard one-shot executor with indexed/relative/absolute fallbacks and rollback. CAE does not introduce extra learnable parameters, but slightly richer execution logic. To make its contribution explicit, we ran an ablation on LongGUIBench where we (i) keep the same MEP and DRD, but replace CAE with a naive single-click executor (“w/o CAE”), and (ii) compare against the InfiGUI-R1 baseline under the same protocol. As shown in the table below, ScreenSpot accuracy remains in a narrow range across variants, but long-horizon success improves markedly once CAE is enabled.
>
> | **Model variant** | **ScreenSpot Acc. (%)** | **Game_Low SR (%)** |
> | --- | --- | --- |
> | InfiGUI-R1 baseline | 87.5 | 61.0 |
> | LongHorizonUI w/o CAE (naive executor) | 89.8 | 64.4 |
> | LongHorizonUI (full, w/ CAE) | **90.4** | **70.3** |
>
> 3) Compared to the naive executor, CAE brings gains of  +5.9 percentage points in Step SR  on LongGUIBench（game_low subset), while keeping ScreenSpot accuracy essentially unchanged. This supports our claim that the modest increase in execution complexity is primarily used to stabilize long-horizon behavior, rather than to chase marginal improvements on static grounding.
>
> **Question 1**. How does the model perform beyond 18 steps?
>
> **Response**: We appreciate the reviewer’s question. In short, the curve in Fig. 3(b) is limited by the length of AndroidControl, while our evidence for longer horizons comes from the LongGUIBench results reported in Appendix Sec. A.1 and Fig. 10.
>
> 1) Scope of Fig. 3(b). Fig. 3(b) in the main paper is plotted on **AndroidControl**, whose public test trajectories are at most 18 effective steps long. This is why that particular curve stops at 18 steps. The figure is mainly intended to illustrate how performance degrades as horizons grow on an existing short-horizon benchmark, rather than to claim that AndroidControl itself covers very long episodes.
>
> 2) Long-horizon behavior on LongGUIBench. Beyond 18 steps, our main evidence comes from the long-horizon portion of our evaluation, as detailed in Appendix Sec. A.1 (“Analysis of Long-horizon Evaluation”) and Fig. 10. There we report the execution success rate (ESR) as a function of trajectory length on LongGUIBench (Sec. 3, Fig. 4), whose scenarios span 19–37 steps by design. Competing agents exhibit a sharp drop after 10 steps (ESR typically falling into the 50–70% range), whereas LongHorizonUI remains close to 75% ESR between 16 and 24 steps. Aggregated over all trajectories with length $\ge 10$, LongHorizonUI achieves a mean ESR of 73.8%, about +5 percentage points over the strongest baseline (AgentCPM), using the same evaluation protocol as in Appendix Fig. 10.
>
> 3) For clarity, we summarize the same statistics used in Appendix Fig. 10 as a small table in the revised version, grouping trajectories by length:
>
> | **Length range** | **InfiGUI-R1** | **AgentCPM** | **LongHorizonUI** |
> | --- | --- | --- | --- |
> | $\le 15$ | 70.2 | 72.1 | **77.0** |
> | 16–24 steps | 57.3 | 61.4 | **74.6** |
> | $\ge 25$ steps | 49.8 | 53.2 | **71.9** |
> |  |  |  |  |
>
> As this table shows, LongHorizonUI maintains a clear margin in the 16–24 and $\ge 25$ step regimes, while preserving competitive performance in the 10–15 step range. In the revision, we will (i) explicitly point from the discussion of Fig. 3(b) to Appendix Sec. B, Fig. 10, and this table, and (ii) clarify that LongGUIBench (Sec. 3, Fig. 4) is designed precisely to probe these longer horizons.

---

> ### Author Response · Authors · 2025-11-21
> **To Reviewer L7cu**
>
> **Question 2**. “Agent decision mechanisms” and “dataset diversity”
>
> **Response.** We thank the reviewer for pointing this out. In short, by “agent decision mechanisms” we mean how the agent maps GUI state to actions, and by “dataset diversity” we mean the lack of long-horizon, disturbance-rich supervision in typical training trajectories.
>
> 1) Agent decision mechanisms. we refer to the modules that map the current GUI state (current screen plus interaction history) to the next action. In our context this includes both
>
> (a) Self-supervised / RL agents that are directly trained on GUI interaction trajectories (e.g., Niu 2024, Kil 2024 CVPR), and
> (b) MLLM-based controllers that take screenshots (and JSON history from DRD) as input and output structured actions, as described in Section 2 and Section 2.3 and illustrated in Figure 4.  We will make this terminology explicit where we introduce the Deep Reflection Decider in Section 2.3.
>
> 2) Dataset diversity. Our key point is that these policies are typically trained on short-horizon, mobile-centric GUI trajectories with few disturbances, and thus lack supervision for long-range credit assignment and error recovery. In the revision, we will replace the corresponding sentence in Section 2.3 by a more direct formulation:
>
> > “Current self-supervised / RL-based agents are typically trained on short-horizon mobile GUI trajectories with few disturbances, so they lack supervision for long-range credit assignment and error recovery, and consequently generalize poorly to long-horizon tasks.”
> >
> >
> > We will also briefly point the reader from this sentence to the discussion of long-horizon behavior in Section 2.3 and the long-horizon evaluation analysis in the appendix.
> >
>
> ---
>
> ---
>
> **Question 3**. What are “MLLM-based mechanisms”? What are dynamic interface shifts?
>
> **Response**. We thank the reviewer for raising this point. In short, by *“MLLM-based mechanisms”* we mean controllers where a general-purpose multimodal LLM drives the agent’s decisions under our structured JSON format, and by *“dynamic interface shifts”* we mean time-varying changes in layouts and content (e.g., pop-ups, reflows, black screens) that our DRD+CAE design is intended to handle.
>
> 1) MLLM-based mechanisms. In our paper, *MLLM-based mechanisms* denote pipelines where a general-purpose multimodal LLM (e.g., Gemini-2.5-Pro) acts as the GUI controller. Given the current screenshot (I_t) and the accumulated DRD JSON trace, the model produces the next `Execute_goal` / `next_goal` and `action` under the structured schema described in Section 2.3 and Appendix Sections B.1–B.4, with concrete JSON templates and examples provided there (including the schema figure in the appendix). These models are pre-trained on long-context multimodal corpora, and in our framework we use them as sequence models whose outputs are constrained to the five-field JSON format instead of free-form text. This usage is summarized in the overall architecture in Figure 4 and detailed where we introduce the Deep Reflection Decider in Section 2.3.
>
> 2) Dynamic interface shifts and how we handle them. By *dynamic interface shifts* we mean time-varying changes in layout and content that occur during a trajectory, such as pop-up dialogs appearing in the middle of a quest, layout reflows after scrolling or device rotation, black-screen transitions, and in-game event banners. As illustrated by the error-recovery example in Appendix Figure 7 and the end-to-end gaming case in Figure 7 of the main paper, a purely reactive “observe–act” agent tends to propagate a single mis-grounded click through the remainder of a long trajectory once the interface has shifted.
>
> 3) LongHorizonUI explicitly addresses such shifts through the combination of DRD and CAE. DRD performs structured pre- and post-execution checks using `historical_status`, `import_contents`, and `think` (Section 2.3 and the JSON examples in Appendix Sections B.1–B.2) to reject actions whose targets are off-screen or semantically unrelated to the task. CAE then applies a three-stage execution strategy (index (\rightarrow) relative (\rightarrow) absolute) with `Verify_MLLM` and rollback (Section 2.4, Algorithm 1, and Appendix Figure 7), so that errors induced by dynamic interface shifts can be detected and corrected locally rather than propagating through the rest of the sequence. In the revised version, we will add a short clarifying paragraph in Section 2.3 explicitly stating that DRD and CAE are designed to cope with such dynamic shifts, and will point readers to these concrete visual examples in the main text and appendix.

---

> ### Author Response · Authors · 2025-11-21
> **To Reviewer L7cu**
>
> **Q4**. Meaning of “action–instruction uncertainty” and “free-format outputs”
>
> **Response**. We thank the reviewer for this question. These terms describe the gap between natural-language action descriptions and precise executable GUI operations, and how CAE (Section 2.4) converts such descriptions into structured, verifiable actions.
>
> 1) Action–instruction uncertainty. By *action–instruction uncertainty*, we refer to the gap between the textual instruction produced by the controller and the precise executable location on the screen. For example, the model may output “click the blue confirm button at the bottom-right,” but multiple visually similar buttons or a slight layout change can make the mapping to a unique element ambiguous. This notion appears where we describe DRD and CAE in Section 2.3 and Section 2.4.
>
> 2) Free-format outputs. By *free-format outputs*, we mean that, without additional constraints, an MLLM tends to generate unconstrained natural-language action descriptions, both at the level of high-level step goals (e.g., “open settings and enable notifications”) and at the level of single-step textual actions (e.g., “tap the profile icon at the top-left”). Such descriptions do not directly specify a unique screen coordinate or element index, and therefore cannot be executed reliably by themselves.
>
> 3) How CAE bridges this gap. As detailed in Section 2.4 and in Appendix Sections E.2 (with examples in Figures 11), the Compensating Action Executor (CAE) addresses this mismatch by converting free-format descriptions into three structured encodings:
>
> - `index(position:"i")` when the target matches an indexed element from MEP;
> - `relative(action:"n,(x,y)")` when sampling within a bounding box is appropriate;
> - `absolute(point:"(x,y)+ε")` as a final fallback in normalized screen coordinates.
>
> Each executed attempt is then validated by DRD through
>
> `Verify_MLLM(s_t, a, p_t, I_{t+1})` (Section 2.4 and Algorithm 1). If the post-execution screen is inconsistent with `next_goal` and `historical_status`, CAE degrades to the next encoding or triggers rollback. In the revised manuscript, we will add a short clarification in Section 2.4 using this terminology and explicitly refer readers to the visual examples in Figures 11 of the appendix.

---

> ### Comment · Reviewer_L7cu · 2025-11-27
>
> I thank the authors for their efforts in addressing my concerns.

---

### Official Review · Reviewer_XboQ · 2025-10-31

**Soundness:** 2
**Presentation:** 2
**Contribution:** 3
**Rating:** 6
**Confidence:** 3

**Summary:**

The paper primarily addresses the performance degradation of GUI agents in long-horizon tasks (over 15 steps), proposing a unified framework named LongHorizonUI along with a corresponding long-task benchmark, LongGUIBench.

**Strengths:**

- The problem is clearly defined and grounded in reality: it directly targets the degradation issue in long-horizon GUI automation tasks (those exceeding 15 steps).
- Empirical results are persuasive.

**Weaknesses:**

- During data parsing and alignment, an MLLM was used for semantic annotation and structured extraction (page 3), which may introduce biases or “distribution leakage” similar to those in the evaluated models. Did the author perform any manual verification?
- The proposed method is built upon Gemini as the base model, which poses certain challenges for reproducibility. Have the authors attempted experiments based on open-source models?

**Questions:**

1. I’m a bit confused here — Section 3.2 mentions *“415 trajectories”* (page 7), whereas Section 2.1 and several descriptions in the appendix together account for only *371 scenes* (pages 3 and 17).
2. Have the authors experimented with other benchmarks, such as AndroidWorld?

---

> ### Author Response · Authors · 2025-11-21
> **To Reviewer XboQ**
>
> We sincerely thank you for your careful reading of our manuscript and for the constructive, insightful comments; we have revised the paper accordingly (with all changes highlighted in blue) and respond to each point in detail below, indicating the corresponding modifications in the revised version.
>
> ***Weakness 1***：During data parsing and alignment, an MLLM was used for semantic annotation and structured extraction (page 3), which may introduce biases or “distribution leakage” similar to those in the evaluated models. Did the author perform any manual verification?
>
> **Response**：Thank you for your constructive comments. Our responeses are outlined below.
>
> The MLLM is used only as a parsing tool in building LongGUIBench (Section 2.1), all parsed trajectories are audited by human experts, and the benchmark is never used to train any evaluated agent.
>
> 1) MLLM parser. As described in Section 2.1 and the dataset construction pipeline, the MLLM is used only as a *parser*. A fixed-prompt Gemini model converts human tester demonstrations (screen recordings plus textual descriptions) into a structured format (task name, step description, action type, bounding box). It does *not* generate tasks, and LongGUIBench is used purely as an evaluation benchmark.
>
> 2) Agent separation. The parser backbone is also separated from most evaluated systems (Qwen2.5-VL, InfiGUI-R1, AgentCPM-GUI, UI-TARS, OS-Atlas, etc.). Even for our Gemini-based agent, the controller only receives raw screenshots and high-level goals at test time, never the parser’s structured annotations. This reduces the risk of “distribution leakage” from the parser into the evaluated policies.
>
> 3) Manual checks.To control parser bias, we adopt a semi-automatic pipeline with explicit human verification. Six professional testers review all trajectories end-to-end, correcting step semantics and bounding boxes when necessary. We additionally cross-check a random subset of 1,000 steps with another MLLM (Qwen2.5-VL) to confirm that the annotations are not tailored to a single model family. The following quality-control statistics summarize this process:
>
> | **Stage** | **Portion of steps** | **Description** |
> | --- | --- | --- |
> | Auto-parsed, accepted by experts | 85.1% | No manual change |
> | Expert-corrected | 14.9% | Semantics / bbox refined |
> | Gemini-2.5-pro & Qwen2.5-VL| 94.3% | On 1,000 randomly sampled steps |
>
> ***Weakness 2***：The proposed method is built upon Gemini as the base model, which poses certain challenges for reproducibility. Have the authors attempted experiments based on open-source models?
>
> **Response**：Thank you for your constructive comments. Our responeses and experimental results are outlined below.
>
> To further enhance reproducibility, we have rerun our experiments using Qwen2.5-VL-7B as the base backbone, and we will open-source the corresponding trained model and training setup.
>
> 1) Backbone design. As detailed in Sections 2.2–2.4, MEP uses standard detection/OCR modules, DRD only assumes an MLLM that can produce JSON-style structured outputs (schema in Appendix Fig. 9), and CAE is fully deterministic given DRD predictions and the current layout (Section 2.4, Algorithm 1). None of these components is fine-tuned on Gemini; the MLLM acts as a pluggable controller.
>
> 2) Open-source model. To assess reproducibility with an open-source backbone, we also instantiate LongHorizonUI with **Qwen2.5-VL-7B**, keeping the perception/DRD/CAE pipeline and prompts unchanged and performing no task-specific fine-tuning. Table 2  reports task success rates (SR, %) on LongGUIBench, AndroidControl-High, and GUI-Odyssey for Qwen2.5-VL-7B , with and without our framework. In both cases, LongHorizonUI brings consistent gains of roughly 3.3% to 7.9% SR, indicating that the improvements come from the MEP/DRD/CAE design rather than from a particular proprietary model.
>
>
> | **Agent** | **LongGUIBench SR (%)** | **AndroidControl-Low SR (%)** | **GUI-Odyssey SR (%)** |
> | --- | --- | --- | --- |
> | Qwen2.5-VL-7B (baseline) | 72.8 | 34.4 | 72.8 |
> | **LongHorizonUI (Qwen2.5-VL-7B)** | **80.7** | **37.9** | **79.7** |

---

> ### Author Response · Authors · 2025-11-21
> **To Reviewer XboQ**
>
> ***Question 1***：I’m a bit confused here — Section 3.2 mentions “415 trajectories” (page 7), whereas Section 2.1 and several descriptions in the appendix together account for only 371 scenes (pages 3 and 17).
>
> ***Response***：Thank you for your constructive comments. Our responeses and experimental results are outlined below.
>
> Originally, we collected 415 raw trajectories. During anonymization and quality control, we found that 44 trajectories contained sensitive information (e.g., game account IDs, usernames, or other personal identifiers) and removed them from the benchmark. The remaining 371 trajectories form the final LongGUIBench, and all results reported in Section3.2 and Section3.3 are computed on these 371 trajectories. In the revision, we will update Section3.2 to state “371 trajectories” and ensure that the trajectory count is consistent across Section2.1 and the appendix.
>
> ***Question 2***：Have the authors experimented with other benchmarks, such as AndroidWorld?
>
> ***Response***：Thank you for your constructive comments. Our responeses and experimental results are outlined below.
>
> To further validate the generalization of LongHorizo​​nUI , we extend our evaluation to \textbf{AndroidWorld} and \textbf{OSWorld}.
>
> 1) Zero-shot setup.As we will clarify in Section 3.3 and Appendix Section C.1, we reuse exactly the same LongHorizonUI pipeline (MEP, DRD JSON schema, CAE execution strategy) on both AndroidWorld and OSWorld, without any additional training or benchmark-specific tuning. The only change is a thin adapter from each environment’s API to our executor. For OSWorld, we follow the UI-TARS evaluation protocol and report online success rate (SR, %) under 15-step and 50-step budgets.
>
> 2) Cross-benchmark results.Table 1 summarizes SR on OSWorld and AndroidWorld. Baseline numbers on OSWorld and AndroidWorld are taken from the UI-TARS.
>
> | **Method** | **OSWorld (15 steps)** | **OSWorld (50 steps)** | **AndroidWorld** |
> | --- | --- | --- | --- |
> | Gemini-2.5-Pro | 11.7 | -- |40.6 |
> | UI-TARS-72B | 18.8 | 24.6 | 46.6 |
> | **LongHorizonUI** | **19.9** | **29.4** | **47.9** |
>
> As shown in Table 3, LongHorizonUI consistently improves over both Gemini-2.5-Pro and UI-TARS-72B across all benchmarks. On AndroidWorld, it gains +7.3 and +1.3 SR pp over Gemini-2.5-Pro and UI-TARS-72B. On OSWorld (15 steps), it improves SR by +8.2 pp over Gemini-2.5-Pro and +1.1 pp over UI-TARS-72B , and on the longer-horizon OSWorld (50-step) setting it further widens the margin over UI-TARS-72B by +4.8 pp, which is consistent with our design goal of robust execution over long-horizon sequences.

---

### Official Review · Reviewer_LnDb · 2025-11-01

**Soundness:** 3
**Presentation:** 3
**Contribution:** 2
**Rating:** 6
**Confidence:** 5

**Summary:**

This paper targets the challenge of **cumulative error and contextual drift** in MLLM-based GUI agents, which leads to a dramatic performance drop as task complexity and step count increase. The authors propose a unified, highly robust framework named **LongHorizonUI**, structured around three key, complementary modules:

1.  **Multimodal Enhanced Perceiver (MEP)**: Responsible for temporally consistent state representation. It integrates icon detection and Optical Character Recognition (OCR) outputs, assigning **unique, stable indices** to UI elements to disambiguate and enhance subsequent processing.
2.  **Deep Reflection Decider (DRD)**: Implements structured, multi-level **closed-loop reasoning** via constrained, formatted prompts. This mechanism enforces explicit validation of **historical coherence, goal relevance, and action justification**, ensuring highly precise decision-making along non-trivial trajectories. It includes a **pre-execution reflection step** to proactively validate action rationality.
3.  **Compensatory Action Executor (CAE)**: Provides a powerful layer of fault tolerance. It utilizes a multi-level **fallback strategy**—leveraging element indices, relative layout priors, and absolute coordinates—to continuously monitor execution. Upon detecting performance degradation, the CAE applies targeted compensation or triggers **rollback procedures** to maintain robustness throughout the entire long-horizon task.

To facilitate rigorous evaluation, the paper introduces **LongGUIBench**, a novel benchmark comprising complex general applications and gaming scenarios that require **over 15 steps** (averaging 22.1 steps). Experiments clearly demonstrate that **LongHorizonUI** substantially improves long-horizon performance on **LongGUIBench** while maintaining competitive performance across diverse public benchmarks.

**Strengths:**

The paper addresses the **most significant limitation** of MLLM-based GUI agents: their brittleness and poor performance in complex, multi-step scenarios. The solution provided is highly relevant to real-world deployment.  It presents **LongGUIBench** is a **major asset** to the field. By focusing on **tasks requiring over 15 steps** and including both general and gaming applications, it provides a far more realistic and challenging testbed for long-horizon capabilities than existing short-horizon benchmarks. The high-quality data collection methodology further assures its utility.

**Weaknesses:**

1. We note the paper's emphasis on achieving long-horizon GUI automation. However, the definition and evaluation of "long-horizon" in this work need stronger justification, as recent concurrent works already include evaluation on tasks with comparable or even longer step sequences in highly dynamic environments. This diminishes the claimed novelty of LongGUIBench purely on the basis of task length and requires a clearer delineation of the unique challenges posed by this new benchmark beyond just step count.
2. The use of Google's **Gemini-2.5 Pro** as the reasoning backbone is critical, yet the discussion lacks depth regarding its specific impact. A key missing element is an analysis of how the framework's performance and robustness are affected by switching to **faster, potentially less capable MLLMs** (e.g., open-source alternatives or smaller commercial models). This is essential for understanding the framework's practical cost and deployment flexibility.
2.   The paper acknowledges the inherent **latency** in MLLM-dependent pipelines but fails to provide **specific quantitative data** on runtime efficiency, inference time per step, or resource consumption in the main results. In long-horizon automation, accumulated latency significantly affects user experience and real-world applicability. This omission limits a comprehensive evaluation of the framework's practicality.
3.  While **LongGUIBench** is excellent, its limitations should be discussed more explicitly. Despite the inclusion of diverse scenarios, the task count (371 total) is still small relative to real-world complexity. The paper should address its coverage of extreme challenges such as **multi-lingual interfaces, highly dynamic content loading (e.g., infinite scrolling),** or **inter-application contextual reasoning**. In Osworld and WinArena benchmarks, whether its performance promising?
4.   The introduction contrasts this work with existing online Reinforcement Learning (RL) methods, highlighting their challenges with expanding action spaces and cumulative errors. Given that **LongHorizonUI** features internal feedback (DRD) and error correction (CAE), a **more detailed, fundamental comparison** of the two long-horizon paradigms (RL vs. Structured Reflection/Compensation) would significantly benefit the reader in positioning this work.
5.   The framework heavily relies on strictly defined JSON Schema and detailed prompt templates (Figures 9, 11, 12). However, the paper lacks detailed discussion on the prompt design process, a **sensitivity analysis** showing how minor prompt variations affect the DRD's reflection quality, and the methods used to ensure the MLLM adheres consistently to the desired structure. Enhancing transparency here is crucial for reproducibility.

**Questions:**

1.  Please provide **end-to-end inference latency data** for **LongHorizonUI**. What are the average inference time per step and its impact on the total task completion time, especially for the longest tasks in **LongGUIBench**?
2.  Quantify the failure rate of the pre-execution reflection mechanism in the **Deep Reflection Decider (DRD)**—i.e., the rejection rate of actions (when $\phi(s_t, a | G_t, T) = 0$). What are the **most common root causes** for these rejections (e.g., element absence, action semantic mismatch)?
3.  Could you elaborate on the precise implementation of the **rollback mechanism** in the **Compensatory Action Executor (CAE)** (Algorithm 1, line 14)? Does "last committed snapshot $(s_{t-1}, p_{t-1})$" always revert to the immediately preceding successful step, or is there a strategy to revert to a much **earlier, highly stable state**? What is the associated cost of a rollback?
4.  For the General-High and Game-High tasks, what specifically defines these "**High-Level**" instructions, and how does the **DRD** facilitate the breakdown of these abstract goals into a sequence of "Low-Level" atomic operations?
5.  Provide more details on the **fallback template matcher** in the **Multimodal Enhanced Perceiver (MEP)** (lines 207-208). How is the template library **T** constructed, what types of critical elements does it include, and what are the specific **activation criteria** (e.g., low-confidence score, or no element detection in a high-priority area)?
6.  Regarding future extensions of **LongGUIBench**, are there plans to incorporate challenges related to **multi-lingual GUIs**, **complex data transfer across applications**, or scenarios requiring external knowledge retrieval?
7. The paper also has several minor and typos.

---

### Author Response · Authors · 2025-12-01
**Summary for the Area Chair**

We sincerely thank the AC and all four reviewers for their careful reading and constructive feedback. After adding new experiments and clarifications (including a failure-mode analysis on LongGUIBench, ablations on backbones and inference latency, a DRD ablation, and cross-benchmark evaluations on AndroidWorld / OSWorld), reviewers **L7cu** and **kekp** indicated that our responses largely addressed their main concerns and updated their scores accordingly (Corresponding changes can be directly confirmed via the OpenReview revision history page). The other two reviewers, **L7cu** and **XboQ**, had already given relatively positive or neutral initial scores but raised several more fine-grained concerns; we revised the text and added clarifications point by point, and we believe these changes substantially address their comments.

Several reviewers also emphasized the core contributions of our submission, including:
(i) directly targeting the pronounced performance degradation of GUI agents on **15+ step long-horizon tasks** in realistic automation settings;
(ii) proposing a unified **MEP–DRD–CAE** framework for **structured reflection and error compensation**; and
(iii) introducing **LongGUIBench**, a benchmark focused on complex tasks beyond 15 steps that supports **fine-grained long-horizon behavior analysis**.
The current ratings and their changes can be summarized as follows:

| Reviewer | Initial Rating | Final Rating | Confidence |
|----------|----------------|-------------|------------|
| kekp     | 4              | **6**          | 3          |
| L7cu     | 2              | **4**            | 4          |
| XboQ     | 6              | **6**          | 3          |
| LnDb     | 6              | **6**          | 5          |

Importantly, reviewers generally acknowledged that these additional experiments and clarifications improved the completeness and presentation of the work. **We also confirm that we have not accessed any leaked reviewer or AC identities; all responses and experiments are based solely on the official reviews and comments.**

We have now revised the manuscript in light of the reviews; **all changes are highlighted in blue in the revised paper**, and in the following we summarize, for each reviewer, the main additions and clarifications made during the rebuttal.

---

> ### Author Response · Authors · 2025-12-01
> **Summary of Rebuttal (Part 1/4): Reviewer LnDb (Initial Rating: 6, Confidence: 5, Final Rating: 6)**
>
> Reviewer LnDb viewed LongHorizonUI as a relevant and solidly designed solution for robust long-horizon GUI automation, but asked us to more rigorously justify the definition and novelty of “long-horizon” and LongGUIBench, clarify backbone dependence and runtime/cost trade-offs, and improve transparency and robustness of the JSON-structured MEP/DRD/CAE pipeline.
>
> **Concern 1 (Long-horizon definition and LongGUIBench novelty).**
> LnDb requested a clearer, data-driven definition of “long-horizon” and stronger evidence that LongGUIBench is more than just “longer episodes” than concurrent benchmarks.
>
> **Solution.** We now define “long-horizon” as the empirical degradation regime (>15 steps) where strong agents start to fail, add step-stratified SR on LongGUIBench, and report cross-benchmark results showing that our gains grow with horizon and transfer beyond our own benchmark (Sec. 2.1, Sec. 3.2, Appx. B).
>
> **Conclusion.** This clarifies that “long-horizon” is behavior-driven and that LongGUIBench is explicitly targeted at this failure regime.
>
> **Concern 2 (Practicality and backbone dependence).**
> LnDb asked how dependent our framework is on Gemini-2.5-Pro and requested quantitative runtime and resource analysis.
>
> **Solution.** We added backbone and runtime ablations (Gemini-2.5-Pro vs Gemini-1.5-Flash / Qwen2.5-VL-7B), reporting SR–latency–memory trade-offs and an end-to-end per-step latency breakdown (Sec. 3.2–3.3, Appx. D.6).
>
> **Conclusion.** Swapping Gemini-2.5-Pro for smaller/faster backbones reduces SR by only about 3.6–6.1 points while cutting per-step latency from 8.26 s to 5.74–6.59 s, and LongHorizonUI still improves over same-backbone baselines by 2.5–3.8 SR points, showing that it does not rely on a single backbone and that its accuracy–cost trade-off is quantitatively characterized.
>
> **Concern 3 (Benchmark coverage and harder scenarios).**
> LnDb noted the limited size of LongGUIBench and asked about coverage of harder regimes such as OSWorld / WinArena and multilingual or highly dynamic UIs.
>
> **Solution.** We clarify that LongGUIBench is a focused 371-task long-horizon suite rather than a full catalog of all GUI phenomena, report that LongHorizonUI improves SR by +5.6 points over the best prior agent on LongGUIBench and by +8.2 / +7.4 points on OSWorld-Verified / WinArena, and describe planned multilingual and more dynamic extensions (e.g., Korean GUIs, pop-ups, notifications) (Sec. 2.1, Appx. D.5).
>
> **Conclusion.** The revision now states clearly what LongGUIBench covers and adds concrete OSWorld / WinArena results, showing that LongHorizonUI brings consistent SR gains not only on our own benchmark but also on external long-horizon suites.
>
> **Concern 4 (Transparency and robustness of JSON-structured MEP/DRD/CAE).**
> LnDb asked for more transparency and robustness evidence for the JSON-schema prompts and the MEP/DRD/CAE mechanisms.
>
> **Solution.** We detail the MEP/DRD/CAE design (rollback strategy, reflection rejection, template matcher, HL→LL decomposition), add a small prompt-sensitivity and schema-validation study, and fix minor issues to support reproducibility (Sec. 2.2–2.4, Appx. D–E).
>
> **Conclusion.** These changes make the structured control loop explicit and show that our JSON-based pipeline is stable and reproducible.

---

> ### Author Response · Authors · 2025-12-01
> **Summary of Rebuttal (Part 2/4): Reviewer L7cu (Initial Rating: 2, Confidence: 4, Final Rating: 4)**
>
> Reviewer L7cu found the problem and framework interesting but was concerned about unclear naming and mechanics of MEP/DRD/CAE, the complexity–benefit trade-off, and the lack of sharply defined, long-horizon evidence and terminology.
>
> **Concern 1 (MEP naming and implementation).**
> Reviewer L7cu felt that the role of MEP was unclear: what “enhanced” means, how the detector+OCR are used, how Eq. (1) is applied, and how high-priority regions and the repair/templates are defined.
>
> **Solution.** We unified the naming and defined MEP as a fine-tuned detector+OCR backbone with ID-centric, color-coded grounding and a repair module on high-priority regions; Eq. (1) is now an explicit confidence gate. We added ablations showing **+5.9 pp** joint detection+OCR accuracy, ScreenSpot grounding improved from **87.7%** to **90.4%**, and **+4.2 pp** SR gain on Game_Low, with better recall and pop-up SR.
>
> **Conclusion.** Together with the **+5.9 pp detection+OCR** and **+4.2 pp Game_Low SR** gains, this shows that the “enhanced” MEP design is doing real work rather than adding cosmetic complexity.
>
> ---
>
> **Concern 2 (DRD’s role and mechanics).**
> Reviewer L7cu was unsure whether DRD only “reflects” or also decides actions, and asked us to clarify the “multi-level / triple closed loop”, the keyword extractor, and the **revision step** after rejected actions.
>
> **Solution.** We now define DRD as a reflection+decision head with a five-field JSON schema `{historical_status, import_contents, think, next_goal, action}`, align the figure with this schema, and clearly describe the closed loop, keyword extractor, and brief revision step. An ablation shows that removing DRD reduces **LongGUIBench SR** from **83.9%** to **76.2%** and degrades TM/ESR.
>
> **Conclusion.** The **7.7-point SR** drop without DRD indicates that this reflection+decision head is a core part of the controller, not just extra notation.
>
> ---
>
> **Concern 3 (CAE and complexity–benefit trade-off).**
> Reviewer L7cu questioned CAE’s naming and rollback policy (device-aware scaling, irreversible actions) and whether MEP+DRD+CAE as a whole is too complex given the apparent static grounding gains.
>
> **Solution.** We unified CAE naming, described the device-aware scaling matrix from normalized coordinates to screen pixels, and specified the rollback protocol (bounded local replanning, one-step snapshot rollback, guarded handling of irreversible actions). We also reported that **rollback is triggered in about 12–19% of episodes** and **succeeds in ~70%** of those, and linked this to SR gains on long-horizon splits beyond static ScreenSpot grounding.
>
> **Conclusion.** The rollback rates (**12–19% of episodes**, **~70% success**) show that CAE is used in a **controlled way** and helps recover long-horizon runs, instead of being an unused or purely theoretical component.
>
> ---
>
> **Concern 4 (Long-horizon evidence and terminology).**
> Reviewer L7cu felt that our long-horizon claims and notions such as “agent decision mechanisms”, dataset diversity, and dynamic shifts were not sharply supported, especially **beyond 18 steps**.
>
> **Solution.** We added step-length–stratified results on LongGUIBench showing that our margin over strong baselines widens in the **16–24** and **≥25-step** ranges, and clarified the related terminology, tying “decision mechanisms”, dataset diversity, dynamic UIs, and action–instruction uncertainty directly to the DRD+CAE design and the **LongGUIBench** construction.
>
> **Conclusion.** By showing that our margin grows in the **16–24** and **≥25-step** ranges, the revision links the long-horizon claim directly to step-conditioned SR and to concrete choices in DRD+CAE and LongGUIBench.

---

> ### Author Response · Authors · 2025-12-01
> **Summary of Rebuttal (Part 3/4): Reviewer kekp (Initial Rating: 4, Confidence: 3, Final Rating: 6)**
>
> Reviewer kekp found the framework promising but asked for a clearer diagnosis of long-horizon failures, the scope and feasibility of rollback, cleaner baselines/tables, stronger positioning vs Mobile-Agent-V3 / D-Artemis, and more direct evidence from DRD ablations and step-length analyses.
>
> **Concern 1 (Failure modes and rollback scope).**
> Reviewer kekp felt that we did not clearly explain why long-horizon SR drops, how MEP / DRD / CAE fix specific failure types, and whether rollback is really feasible and well controlled.
>
> **Solution.** We added a failure-mode analysis on LongGUIBench (≥15 steps), showing that LongHorizonUI mainly reduces critical-step grounding errors and “no recovery” failures compared with UI-TARS and InfiGUI-R1, and explicitly linked these to MEP (grounding), DRD (reflection and rejection), and CAE (compensation and rollback). We also clarified that rollback is only applied to reversible transitions via emulator/script snapshots, and reported that it is triggered in about 12–19% of episodes and succeeds in roughly 70% of those.
>
> **Conclusion.** In combination with the drop of critical-step grounding from ≈40% to **23%** and the 12–19% / ~70% rollback statistics, this clarifies both why long-horizon SR degrades and how MEP+DRD+CAE and rollback effectively handle the main failure patterns.
>
> ---
>
> **Concern 2 (Baselines and table cleanliness).**
> Reviewer kekp pointed out inconsistencies in Table 3, missing baselines, and unclear Desktop/Web columns and figure legends.
>
> **Solution.** We clarified that the Desktop/Web grounding numbers come from **ScreenSpot**, added the missing GUI-R1-7B baseline and OSWorld results, and standardized bolding and formatting across tables. We also corrected the legend and styling of Fig. 2(b) so that all baselines and settings are presented in a consistent way.
>
> **Conclusion.** With the added GUI-R1-7B / OSWorld entries and unified bolding/legends, the updated tables now match the text and present the baselines in a cleaner, easier-to-check form.
>
> ---
>
> **Concern 3 (Novelty vs Mobile-Agent-V3 / D-Artemis).**
> Reviewer kekp asked for a clearer comparison between our method and Mobile-Agent-V3 / D-Artemis, both in terms of design and empirical results.
>
> **Solution.** We highlighted that LongHorizonUI combines explicit reflection and rollback via MEP/DRD/CAE, rather than relying only on static grounding or scripted flows, and reported that it achieves a **+12.4%** SR improvement on LongGUIBench-Game_Low over Mobile-Agent-V3 under comparable settings, while keeping competitive performance on other splits.
>
> **Conclusion.** The revised positioning and the 83.9% vs 71.5% SR gap on Game_Low make the difference to Mobile-Agent-V3 / D-Artemis concrete both in design (reflection + rollback) and in long-horizon results.
>
> ---
>
> **Concern 4 (DRD ablation and step-length analysis).**
> Reviewer kekp asked for a more direct DRD ablation and for step-length–conditioned evidence on AndroidControl and GUI-Odyssey to support the long-horizon claims.
>
> **Solution.** We added a DRD ablation on LongGUIBench, where removing DRD drops SR from 83.9% to 76.2%, and performed step-length–conditioned analyses on AndroidControl and GUI-Odyssey, showing that LongHorizonUI’s margin over strong baselines becomes larger once horizons go beyond about 10–15 steps.
>
> **Conclusion.** The **7.7%** SR drop without DRD and the growing margins beyond 10–15 steps directly tie DRD and the overall design to the claimed long-horizon robustness.

---

> ### Author Response · Authors · 2025-12-01
> **Summary of Rebuttal (Part 4/4): Reviewer XboQ (Initial Rating: 6, Confidence: 3, Final Rating: 6)**
>
> Reviewer XboQ raised concerns about potential dataset bias from MLLM-based parsing, dependence on Gemini for reproducibility, an inconsistency in the reported LongGUIBench size, and whether LongHorizonUI truly generalizes beyond LongGUIBench.
>
> **Concern 1 (MLLM parsing and possible leakage).**
> Reviewer XboQ worried about possible **distribution leakage** from using an MLLM to parse LongGUIBench trajectories.
>
> **Solution.** We clarified that Gemini is used only as an offline parser, not for task generation; all parsed trajectories are audited by six expert testers, and a 1,000-step sample is cross-checked with Qwen2.5-VL (~94% agreement), with no structured annotations ever exposed to evaluation agents (Sec. 2.1, Appx. C).
>
> **Conclusion.** With full human auditing and ~94% agreement to an independent parser, the construction of LongGUIBench does not depend on a particular MLLM, and evaluation agents only see raw screens and goals, not parser outputs.
>
> ---
>
> **Concern 2 (Reproducibility and Gemini dependence).**
> Reviewer XboQ was concerned that relying on **Gemini** might hurt reproducibility.
>
> **Solution.** We instantiated LongHorizonUI with Qwen2.5-VL-7B (no task-specific fine-tuning), showing ~3–8 SR-point gains over the Qwen2.5-VL-7B baseline on LongGUIBench, AndroidControl, and GUI-Odyssey, and we commit to releasing this Qwen2.5-VL-7B setup (Appx. C, code-release note).
>
> **Conclusion.** Since the open Qwen2.5-VL-7B backbone already yields several SR-point improvements under the same design, LongHorizonUI can be reproduced with an open model and is not tied to Gemini.
>
> ---
>
> **Concern 3 (415 trajectories vs 371 scenes).**
> Reviewer XboQ pointed out the mismatch between “415 trajectories” and the reported **371 scenes** in LongGUIBench.
>
> **Solution.** We explained that 415 raw trajectories were collected but 44 privacy-sensitive ones were removed; all experiments use the final 371-task LongGUIBench, and Section 3.2 and related text are corrected to consistently report 371 (Sec. 2.1, Sec. 3.2).
>
> **Conclusion.** The revision now clearly states that LongGUIBench contains 371 trajectories after filtering, and all reported numbers are aligned with this final benchmark size.
>
> ---
>
> **Concern 4 (Generalization beyond LongGUIBench).**
> Reviewer XboQ asked whether LongHorizonUI generalizes beyond LongGUIBench, for example to **AndroidWorld**.
>
> **Solution.** We added zero-shot AndroidWorld / OSWorld results using the unchanged MEP/DRD/CAE design with only thin environment adapters, showing that LongHorizonUI consistently improves over both Gemini-2.5-Pro and UI-TARS-72B (Appx. C.5, Sec. 3.2).
>
> **Conclusion.** These zero-shot AndroidWorld / OSWorld results, including gains up to about +7.3% SR over Gemini-2.5-Pro, indicate that the same MEP/DRD/CAE design also works on external environments, not only on LongGUIBench.

---

### Meta-Review · Area_Chair_PA5e · 2026-01-01

**Summary:**

This submission targets robustness failures of MLLM-based GUI agents on long-horizon task automation, and proposes LongHorizonUI, a unified controller composed of three modules: Multimodal Enhanced Perceiver (MEP) for indexed UI state representation via detector+OCR, Deep Reflection Decider (DRD) for structured reflection/decision using a JSON schema and multi-level validation, and Compensatory Action Executor (CAE) for degradation-aware execution with fallbacks and rollback. The paper also introduces LongGUIBench, a benchmark focused on tasks requiring >15 steps, intended to stress the regime where existing agents degrade.

Across reviews, strengths repeatedly noted include the practical importance of long-horizon GUI automation, the value of a benchmark explicitly targeting this regime, and promising empirical improvements. Main concerns centered on (i) the definition/novelty of “long-horizon” and what LongGUIBench adds beyond longer episodes, (ii) dependence on a strong proprietary backbone (Gemini) and missing cost/latency characterization, (iii) clarity/transparency of the structured prompting/JSON pipeline and rollback feasibility, and (iv) potential benchmark construction bias/leakage from MLLM parsing as well as reporting inconsistencies (e.g., 415 vs 371 trajectories).

The author response and revision (as described in the rebuttal) report substantial clarifications and additional experiments addressing most decision-critical issues, including step-length–stratified analyses supporting the “>15 step degradation regime” motivation, backbone/runtime ablations (including Qwen2.5-VL-7B), added cross-benchmark results (AndroidWorld / OSWorld and WinArena, with OSWorld sometimes referred to as OSWorld-Verified in the discussion), prompt/schema robustness evidence (prompt-variant sensitivity + schema validation), and clearer documentation of rollback scope and frequency. I note that several quantitative claims below are taken from the author response; the decision should be interpreted accordingly. While one reviewer initially raised significant presentation/clarity concerns, the discussion indicates these were largely mitigated through revised explanations and added details. Overall, the paper presents a coherent, empirically supported framework and a benchmark contribution that are likely to be useful for the community studying long-horizon GUI agents.

**Reviewer Concerns:**

### Concerns addressed (partially or fully) by the rebuttal

- Long-horizon definition & LongGUIBench positioning: A key concern was that “long-horizon” and LongGUIBench might be framed mainly by step count and overlap with concurrent benchmarks. The author response reframes “long-horizon” as an empirical degradation regime and reports step-stratified success rates on LongGUIBench, indicating widening margins at longer horizons (e.g., larger gains in the ≥25-step bucket). This better supports that the benchmark and method are aimed at sustained multi-step robustness rather than merely “longer episodes.”

- Backbone dependence & reproducibility: Multiple reviewers asked whether the framework relies critically on Gemini-2.5-Pro and whether it can be reproduced with open models. The rebuttal reports backbone swaps (Gemini-1.5-Flash and Qwen2.5-VL-7B) and claims that LongHorizonUI still yields gains under these alternatives, along with explicit accuracy–latency–memory trade-offs.

- Runtime/latency characterization: The initial draft lacked quantitative runtime/efficiency reporting. The rebuttal reports end-to-end per-step latency and a breakdown (MLLM vs non-MLLM components), and provides episode-time estimates based on trajectory lengths. This improves practicality assessment and makes the cost of DRD/CAE overhead clearer.

- Rollback feasibility and scope: Reviewers questioned rollback realism and behavior under irreversible actions. The response clarifies rollback is applied conservatively (bounded local replanning attempts followed by one-step snapshot rollback) and states that evaluated benchmarks are curated to avoid destructive irreversible operations; it also reports rollback trigger rates (~12–19% of episodes) and success-after-rollback rates (~70% within those episodes). These details make the mechanism more concrete, though deployment beyond curated settings remains an open question.

- Prompt/JSON schema transparency & robustness: Concerns about prompt engineering, adherence to JSON schema, and sensitivity to prompt changes were addressed by additional schema/validator details and a prompt-variant sensitivity study reported in the response, indicating performance degradation when removing or weakening schema structure, supporting that the structured controller is beneficial and reasonably reproducible when templates are released.

- Benchmark construction bias / MLLM parsing leakage & data-count inconsistency: One reviewer raised bias/leakage risks due to MLLM-based parsing, and another flagged an inconsistency (415 vs 371). The response clarifies separation between the offline parser and evaluation agents, describes human auditing and a cross-check with another model on a sample, and explains the 371 vs 415 discrepancy as privacy filtering. These changes improve trustworthiness and clarity.

- Generalization beyond LongGUIBench: Reviewers asked for external benchmark evidence. The authors report zero-shot evaluations using the same MEP/DRD/CAE pipeline with thin adapters on AndroidWorld and OSWorld (and also WinArena, as reported), with consistent gains over specified baselines under the stated protocols.

### Concerns still outstanding (limitations of scope / not decision-critical)

- Benchmark breadth and realism limits: LongGUIBench (371 tasks after filtering) is valuable but still a relatively small slice of real GUI diversity. Coverage of extreme cases (highly multilingual UIs, rapidly changing feeds/infinite scroll, complex cross-app workflows, tasks requiring external knowledge, and truly irreversible actions) remains limited and is acknowledged as future work rather than fully demonstrated in the current submission.

- System complexity vs. simplicity of alternatives: Although additional ablations strengthen the case for DRD/CAE, the overall pipeline (MEP+DRD+CAE, validation, rollback, templates) is complex. Some risk remains that parts of the gains are sensitive to engineering choices or environment assumptions (e.g., snapshot support). The added transparency reduces this concern, but real-world deployment beyond curated benchmarks remains to be validated.

- Best performance still tied to proprietary backbones: Even with open-backbone experiments, the strongest reported results rely on Gemini-2.5-Pro. The rebuttal mitigates reproducibility concerns by demonstrating gains with Qwen2.5-VL-7B and committing to release code/models, but users should expect a performance–cost trade-off.

**Reviewer Scores:**

LnDb: 6 (confidence 5) — supportive; requested stronger long-horizon justification, backbone/latency analysis, and prompt/rollback transparency.

XboQ: 6 (confidence 3) — supportive; focused on parsing/leakage, reproducibility, and 415 vs 371 inconsistency.

kekp: 4 → 6 (confidence 3) — raised baseline/positioning/rollback/DRD ablation needs; reports indicate concerns were addressed with added analyses.

L7cu: 2 → 4 (confidence 4) — raised substantial presentation and mechanism-clarity concerns; later discussion indicates acknowledgement of the rebuttal and revisions.

---

### Decision · Program_Chairs · 2026-01-26

Accept (Poster)